Resource
# Comparative membrane proteomics reveals diverse cell regulators concentrated at the nuclear envelope

Li-Chun Cheng*, Xi Zhang*, Sabyasachi Baboo* , Julie A Nguyen, Salvador Martinez-Bartolomé, Esther Loose, Jolene Diedrich, John R Yates III , Larry Gerace

**The nuclear envelope (NE) is a subdomain of the ER with prominent roles in nuclear organization, which are largely mediated by its distinctive protein composition. We developed methods to reveal low-abundance transmembrane (TM) proteins concentrated at the NE relative to the peripheral ER. Using label-free proteomics that compared isolated NEs with cytoplasmic membranes, we first identified proteins with apparent NE enrichment. In subsequent authentication, ectopically expressed candidates were analyzed by immunofluorescence microscopy to quantify their targeting to the NE in cultured cells. Ten proteins from a validation set were found to associate preferentially with the NE, including oxidoreductases, enzymes for lipid biosynthesis, and regulators of cell growth and survival. We determined that one of the validated candidates, the palmitoyltransferase Zdhhc6, modifies the NE oxidoreductase Tmx4 and thereby modulates its NE levels. This provides a functional rationale for the NE concentration of Zdhhc6. Overall, our methodology has revealed a group of previously unrecognized proteins concentrated at the NE and additional candidates. Future analysis of these can potentially unveil new mechanistic pathways associated with the NE.**

## Introduction

The nuclear envelope (NE) is a specialized subdomain of the ER that forms the boundary of the nucleus and compartmentalizes chromosomes and associated metabolisms (Dultz & Ellenberg, 2007). It is a double-membrane sheet comprising the outer nuclear membrane (ONM), the inner nuclear membrane (INM), and the nuclear pore membrane connecting these (Obara et al, 2022). The pore membrane is juxtaposed to nuclear pore complexes (NPCs), transport channels spanning the NE that link the nucleus and cytoplasm (Knockenhauer & Schwartz, 2016; Beck & Hurt, 2017; Lin & Hoelz, 2019). NPCs are massive supramolecular assemblies (~100

MDa in higher eukaryotes) containing multiple copies of ~30 different proteins termed nucleoporins ("Nups"). NPCs mediate nucleocytoplasmic trafficking of most macromolecules by facilitated mechanisms involving shuttling nuclear transport receptors that bind to their cargoes and specific Nups (Wing et al, 2022). NPCs also act as significant barriers to passive diffusion of molecules larger than ~20–40 kD (Knockenhauer & Schwartz, 2016; Beck & Hurt, 2017).

The ONM is contiguous with the more peripheral ER and shares biochemical and functional properties with the latter, whereas the INM has nucleus-centered functions that are specified by its distinctive protein composition (Dultz & Ellenberg, 2007; Pawar & Kutay, 2021). In higher eukaryotes, the INM is lined by a polymeric meshwork of nuclear lamins, type V intermediate filament proteins (Burke & Stewart, 2013; Gruenbaum & Foisner, 2015). Three major lamin subtypes with discrete developmental expression profiles are found in vertebrates: lamins A/C, lamin B1, and lamin B2. Moreover, all eukaryotic cells contain a set of transmembrane (TM) proteins concentrated at the INM and at the nuclear pore membrane (Schirmer et al, 2003; Cheng et al, 2019; Pawar & Kutay, 2021). Collectively, nuclear lamins and these TM proteins have critical roles in the nuclear structure and mechanics (Cho et al, 2017; Maurer & Lammerding, 2019; Miroshnikova & Wickstrom, 2022), chromosome organization and maintenance (Kim et al, 2019; Hildebrand & Dekker, 2020), and regulation of signaling and gene expression (Choi & Worman, 2014; Gerace & Tapia, 2018). At least 15 human diseases arise from mutations in lamins and TM proteins of the NE, underscoring their functional significance (Wong & Stewart, 2020; Shin & Worman, 2021).

TM proteins of the INM become membrane-integrated in the peripheral ER during their synthesis. In higher eukaryotes, most of these proteins appear to accumulate at the NE by a diffusion–retention mechanism. This involves diffusive movement of the proteins in the lipid bilayer around NPCs, coupled with their binding to lamins, chromatin, and/or other INM-associated components (Katta et al, 2014; Ungricht & Kutay, 2015). With this mechanism, exchange of TM proteins between ONM and INM is bidirectional and is limited by the size of their cytoplasmic domains (Katta et al, 2014;

---

Department of Molecular Medicine, Scripps Research, La Jolla, CA, USA

Correspondence: jyates@scripps.edu; lgerace@scripps.edu
*Li-Chun Cheng, Xi Zhang, and Sabyasachi Baboo contributed equally to this work

Ungricht & Kutay, 2015). By contrast, in yeast, TM proteins commonly are transported to the INM by receptor-dependent facilitated diffusion around the NPC (King et al, 2006; Meinema et al, 2011), a mechanism that also contributes to INM targeting of some proteins in higher eukaryotes (Mudumbi et al, 2020). The extent to which specific TM proteins accumulate at the NE relative to the peripheral ER is not fixed, but rather, can depend on the cell type (Malik et al, 2010) and its specific functional state (Le et al, 2016).

Most TM proteins concentrated at the NE are known to have specific functions at the INM and/or the NPC (Dultz & Ellenberg, 2007; Pawar & Kutay, 2021). Predicated on this logic, identification of novel NE-enriched proteins provides a framework to deepen an understanding of NE functions. To define such proteins, a comparative or "subtractive" proteomics approach has been deployed (Schirmer et al, 2003; Korfali et al, 2010; Wilkie et al, 2011; Tang et al, 2020), wherein isolated NEs are analyzed in tandem with purified microsomal membranes derived mainly from the peripheral ER, where most TM protein synthesis occurs. Proteins that are detected at higher levels in NEs than in microsomes by proteomics are candidate NE-concentrated proteins. However, this assignment must be independently confirmed by immunolocalization and/or other methods.

These and allied approaches have identified a cohort of TM proteins concentrated at the NE, most of which are relatively abundant, that is, present at more than ~50–100,000 copies per nucleus (Schirmer et al, 2003; Cheng et al, 2019; Pawar & Kutay, 2021). However, the identification of low abundance NE-enriched proteins by these methods has been confounded by several technical issues. First, proteomics datasets reveal that the NE fractions contain well-defined TM markers of cytoplasmic organelles other than the ER, including plasma membrane, Golgi, and mitochondria, indicating their partial co-fractionation with NEs (Schirmer et al, 2003; Korfali et al, 2010; Wilkie et al, 2011; Tang et al, 2020). Because non-ER cytoplasmic membranes are highly under-represented in the isolated microsomes used for comparative filtering (Schirmer et al, 2003; Korfali et al, 2010; Wilkie et al, 2011; Tang et al, 2020), some uncharacterized proteins that preferentially appear in the NE fraction may actually derive from other cytoplasmic organelles. Second, validation of NE-targeting of candidates has relied on ectopic overexpression (Schirmer et al, 2003; Korfali et al, 2010; Malik et al, 2010; Wilkie et al, 2011; Tang et al, 2020), often by transient transfection with chemical reagents. This can result in the accumulation of ectopic proteins in nonphysiological cellular aggregates that are often juxtanuclear (Tapia & Gerace, 2016), a pattern that can be confused with NE localization. Finally, in most cases, quantitative evaluation of ectopic protein localization at the NE versus other cytoplasmic membranes has not been rigorously implemented.

To circumvent these limitations, we developed modified proteomics-based methods for identification of low-abundance proteins concentrated at the NE. Using chemically extracted membranes to enrich TM proteins and thereby increase proteomics depth, candidates were identified by comparing isolated NEs with composite cytoplasmic membrane fractions rather than ER-biased microsomes. Subsequently, a cohort of TM domain-containing candidates enriched in the NE fraction was tested for selective targeting to the NE by low-level ectopic expression in cultured cells

and systematic quantification by immunofluorescence microscopy. We focused on non-abundant cell regulators not previously linked to the NE. We found that most of these candidates showed clear enrichment at the NE by immunofluorescence microscopy when expressed ectopically, confirming the NE association suggested by comparative proteomics.

We initiated a functional analysis of one of the low-abundance NE-enriched proteins identified in our screen, the palmitoyl-transferase Zdhhc6 (Lakkaraju et al, 2012). We determined that a major NE palmitoylation target of Zdhhc6 is the INM-localized oxidoreductase Tmx4 (Cheng et al, 2019). Our analysis of palmitoylation-deficient mutants of Tmx4 suggested that Zdhhc6 modulates the NE concentration of Tmx4, providing a functional rationale for the observed NE enrichment of Zdhhc6. Altogether, our methodology has revealed a cohort of previously unrecognized NE-enriched proteins and numerous additional candidates, many with interesting functional annotations not previously connected to the NE. This provides a robust framework for hypothesis development and testing to augment current functional understanding of the NE.

# Results

### Membrane proteomics to identify candidate NE-concentrated proteins

We sought to develop improved procedures to predict and validate TM proteins concentrated at the NE based on comparative proteomics with isolated NEs (Schirmer et al, 2003; Malik et al, 2010). Because most well-characterized NE proteins are relatively abundant (>50–100,000 copies per nucleus), we wished in particular to uncover low-abundance NE proteins. For this analysis, we analyzed membrane fractions isolated from C3H10T1/2 murine mesenchymal stem cells (designated "C3H" cells below), a model for human diseases linked to the NE (Worman et al, 2010). A major limitation of previous NE proteomics involved imprecise exclusion of proteins from cytoplasmic membranes that co-fractionated with the NE (see the Introduction section) (Schirmer et al, 2003; Korfali et al, 2010; Wilkie et al, 2011; Tang et al, 2020). Therefore, we used membrane fractions comprising an ensemble of all cytoplasmic organelles, instead of ER-biased microsomal membranes, to filter out "background" (Fig 1A).

Starting with a cell homogenate obtained by hypotonic cell lysis (sample 1), we prepared a low-speed nuclear pellet (sample 3), which also contained a substantial amount of trapped peripheral ER (~50% of the total calnexin) and other cytoplasmic membranes. The postnuclear supernatant from this step (sample 2) was fractionated by flotation on a sucrose step gradient to yield light cytoplasmic membrane (LCM, sample 4) and heavy cytoplasmic membrane (HCM, sample 5) fractions. In parallel, the resuspended nuclear pellet (sample 3) was fractionated on a sucrose step gradient to yield a composite LCM/HCM fraction (designated HCM2, sample 6) and isolated nuclei (sample 7). The nuclear fraction then was processed by nuclease digestion and high salt extraction (Cheng et al, 2019), yielding isolated NEs and a "nuclear contents" fraction (samples 8 and 9).

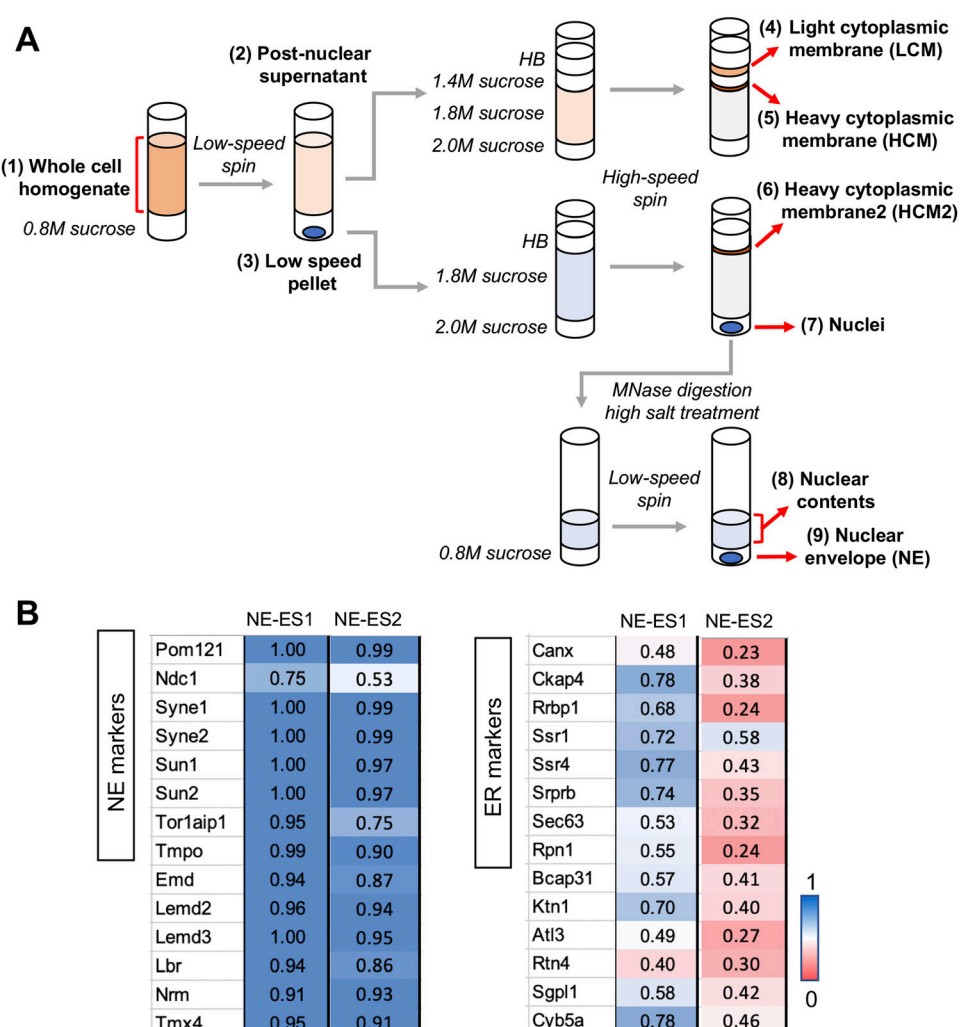

**Figure 1. Isolation and evaluation of membrane fractions used for comparative proteomics.**

**(A)** Schematic diagram depicting isolation of membrane fractions. The whole-cell homogenate from C3H cells (1) was layered over a cushion containing 0.8 M sucrose and centrifuged at low speed to yield a postnuclear supernatant (2) and a low-speed pellet (3). Top: sample (2) was adjusted to 1.8 M sucrose and introduced into a discontinuous sucrose gradient, above a layer containing 2.0 M sucrose, and below a layer with 1.4 M sucrose. The latter was overlaid with a zone of homogenization buffer (HB). Ultracentrifugation resulted in appearance of a large opaque zone at the HB/1.4 M sucrose interface (LCM) and a smaller opaque zone at the 1.4 M sucrose/1.8 M sucrose interface (HCM). Very little material was seen at the 1.8 M sucrose/2.0 M sucrose interface. The LCM and HCM fractions were harvested for subsequent analysis. Middle: sample (3) was resuspended in a buffer containing 1.8 M sucrose and introduced into a discontinuous gradient above a layer containing 2.0 M sucrose and below a layer containing HB. Ultracentrifugation yielded a pellet of isolated nuclei (7), and an opaque zone at the HB/1.8 M sucrose interface (6), which was harvested and designated HCM2. Bottom: sample (7) was resuspended in HB, digested with micrococcal nuclease, adjusted to 0.5 M NaCl, and centrifuged at low speed over a cushion containing 0.8 M sucrose. The pellet from this step (9) comprised the NE fraction and the material above the sucrose shelf was "nuclear contents" (8). **(B)** Evaluation of NE-enrichment scores (NE-ES) in the NE fraction for marker TM proteins of the NE and ER. Values compare less stringent (NE–ES1) and more stringent (NE–ES2) calculation methods, as described in the text. Left panel, NE markers; right panel, ER markers. Ckap4, Rrbp1, Ssr1/4, and Ktn1 are enriched in sheet ER, whereas Atl3 and Rtn4 are more concentrated in tubular ER (see text).

Our previous analysis of isolated NEs, nuclear contents, and other membrane fractions was unable to detect most low-abundance membrane proteins because of the complexity of the fractions (Cheng et al, 2019). We therefore decided to focus on TM proteins, which can be enriched by chemical extraction to increase proteomics depth. We compared the effects of two chemical perturbants known to preferentially deplete peripheral membrane proteins and cytoskeletal elements from membranes (Fig 2A), 0.1 M $Na_2CO_3$ pH 11.5 (Fujiki et al, 1982), and 6 M urea (Steck & Yu, 1973; Foisner & Gerace, 1993). Initially, we used Western blotting to monitor marker proteins in the NE and LCM fractions treated with these conditions (Fig 2B). In NEs extracted with 0.1 M $Na_2CO_3$ (Fig 2B, carbonate wash [Cw]), the peripheral membrane protein lamin A and the cytoskeletal protein actin were efficiently removed as compared with unextracted membranes, whereas the NE-enriched TM proteins LAP2$\beta$ and emerin (Cheng et al, 2019) were not substantially depleted. Similar results were obtained by extraction of NEs with 6 M urea (Fig 2B, urea wash [Uw]), although emerin was partially extracted with this condition. In LCMs, there was no

detectable difference in the extraction resistance of characteristic TM proteins of the mitochondria (Tim23) and ER (calnexin) or in the loss of actin (Fig 2B) with both chemical conditions.

We further compared these two conditions by analyzing NE and LCM fractions by multi-dimensional protein identification technology (MudPIT) (Washburn et al, 2001), involving label-free LC/MS/MS (Table S1). We used normalized spectral abundance factor (NSAF) values to semiquantitatively represent relative protein abundance (see the Materials and Methods section) (Zybailov et al, 2005). For each protein, the NSAF ratios in chemically extracted versus unextracted membranes for the NE (Fig 2C, left panel) and LCM (Fig 2C, right panel) fractions described the enrichment obtained with each chemical condition, and enabled comparison of the two methods (Tables S1 and S2). In a graphical depiction (Fig 2C), proteins whose relative abundance increased with both chemical extraction conditions appeared in the upper-right quadrant, and comprised of mainly TM proteins. Conversely, proteins with diminished abundance in both conditions were in the lower-left quadrant and comprised of mostly non-TM proteins. Both

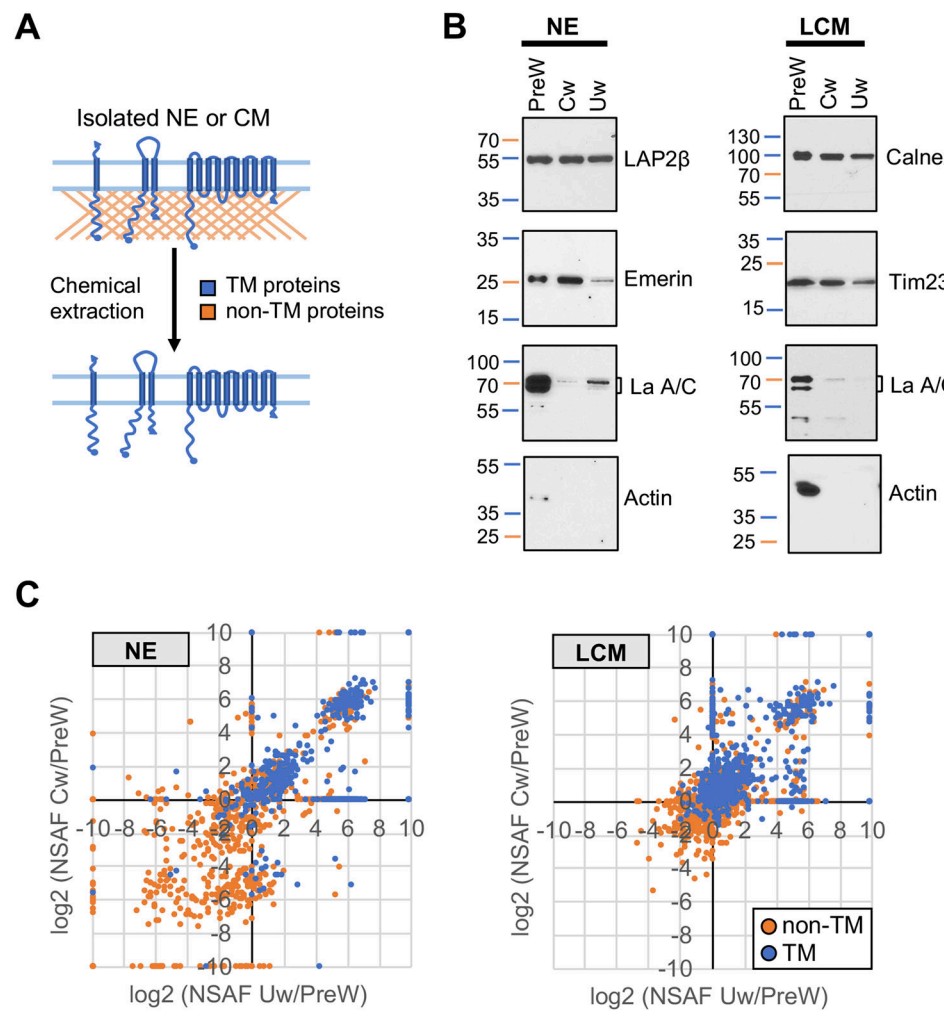

**Figure 2. Evaluation of chemical extraction conditions for TM protein enrichment.** **(A)** Schematic diagram depicting the preferential solubilization of proteins lacking TM segments from membranes that are achieved with 6 M urea or 0.1 M $Na_2CO_3$. **(B)** Western blot analysis of SDS gels to evaluate the presence of characteristic proteins (labeled on right of gels) in NE and LCM fractions. The experiment compared membranes without chemical extraction (PreW) with membranes after treatment with carbonate (Cw) or urea (Uw). Mature lamin A (~72 kD) is not resolved from lamin C (~65 kD) in the NE "preW" sample. Migration of molecular weight markers is indicated on the left of gels. **(C)** Graphs comparing NSAF values of non-TM (brown symbols) and TM proteins (blue symbols) in NE and CML fractions in urea-treated (Uw) or carbonated-treated (Cw) membranes, as compared with their abundance without extraction (PreW). UniProtKB designations were used to define TM proteins. In the analyses in (B, C), NE and LCM membrane fractions from a particular preparation were divided into three aliquots; one was used for PreW analysis, one for Uw, and one for Cw.

extraction conditions enriched TM proteins with strong preference over non-TM proteins. In the NE fraction (Fig 2C, left), more TM proteins showed enrichment by urea but not carbonate extraction (lower-right quadrant), as compared with enrichment by carbonate but not urea extraction (upper-left quadrant). Thus, extraction of membranes with 6 M urea overall seemed to be somewhat better than with 0.1 M $Na_2CO_3$ for efficient recovery of TM proteins. Moreover, because more efficient yields of digested peptides were obtained from samples extracted with urea as compared with carbonate (Table S1), urea extraction was deployed for subsequent analyses.

Altogether, we carried out proteomics of NE and LCM fractions from four independent membrane preparations, analyzing samples extracted with urea from all four preparations (Table S1). In addition, we examined HCM and HCM2 fractions that were extracted with urea for three of the four preparations. To estimate protein abundance in NEs relative to other membrane fractions, we deployed composite NSAF data to compute NE enrichment scores (NE-ES) (Table S3). One metric, termed NE-enrichment score 1 (NE-ES1), incorporated all comparable proteomics data that were available for the four preparations. This involved NE and LCM

fractions that were extracted with urea or carbonate or were not extracted (Table S3). For each protein, NE–ES1 is the ratio of the summated NSAF values from the NE fractions of these samples, compared with summated NSAF values from the (NE + LCM) fractions. We also calculated a second, more stringent, NE-enrichment score 2 (NE–ES2) for the three preparations extracted with urea, for which HCM and HCM2 data also were obtained. NE–ES2 is the ratio of summated NSAF values from the NE fractions, compared to the combined NSAF values for all fractions analyzed (NE, LCM, HCM, and HCM2).

We evaluated the NE–ES1 and NE–ES2-scoring methods by considering abundant and well-characterized TM proteins of the NE and ER (Fig 1B, see annotations in Table S3). For the NE markers, almost all proteins with the exception of Ndc1 had NE–ES1 values >0.9. The NE–ES2 values for these proteins were similarly high, reflecting the minimal detection of these components in the HCM fractions. The relatively low NE–ES values for Ndc1 (0.75, NE–ES1; 0.53, NE–ES2) contrasted with the high values seen for two other NPC-associated TM proteins, Pom121 and Smpd4 (NE–ES1 and NE–ES2, 0.98–1; Table S3). These results, together with data from differentiated adipocytes (Cheng et al, 2019), suggest that there may

**Table 1.  Proteins analyzed by immunofluorescence targeting assay.**

| Protein | Description | NE-ES1 | SPC[a] NE-ES1 | NE-ES2 | SPC[a] NE-ES2 |
|---|---|---|---|---|---|
| Lipid biosynthesis and modification | | | | | |
| Cdipt | CDP-diacylglycerol–inositol 3-phosphatidyltransferase | 1.0 | 130 | 0.93 | 71 |
| Chpt1 | Cholinephosphotransferase 1 | 1.0 | 42 | 0.94 | 29 |
| Gpat4 | Glycerol-3-phosphate acyltransferase 4 | 0.7 | 65 | 0.94 | 32 |
| Ptdss2 | Phosphatidylserine synthase 2 | 1.0 | 99 | 1.0 | 45 |
| Zdhhc6 | Palmitoyltransferase ZDHHC6 | 1.0 | 66 | 1.0 | 30 |
| Oxidoreductases | | | | | |
| Hsd11b1 | Corticosteroid 11-beta-dehydrogenase isozyme 1 | 1.0 | 49 | 1.0 | 6 |
| Ogfod3 | 2-oxoglutarate and iron-dependent oxygenase domain-containing protein 3 | 1.0 | 47 | 1.0 | 20 |
| Vkorc1l1 | Vitamin K epoxide reductase complex subunit 1-like protein 1 | 1.0 | 86 | 0.91 | 55 |
| Signaling regulators | | | | | |
| Tmbim6 | Transmembrane BAX Inhibitor Motif Containing 6 | 0.76 | 113 | 0.65 | 88 |
| Tmem53 | Transmembrane protein 53 | 1.0 | 24 | 1.0 | 10 |
| Tmem161a | Transmembrane protein 161A | 1.0 | 45 | 0.82 | 34 |
| Tmem214 | Transmembrane protein 214 | 0.84 | 831 | 0.74 | 495 |
| Protein folding, misc. | | | | | |
| Dnajc16 | DnaJ Heat Shock Protein Family (Hsp40) Member C16 | 0.93 | 44 | 0.79 | 22 |
| Pigb | GPI mannosyltransferase 3 | 1.0 | 41 | 0.93 | 25 |
| Tmem9 | Transmembrane protein 9 | 1.0 | 90 | 1.0 | 32 |
| Tmem209 | Transmembrane protein 209 | 0.94 | 821 | 0.86 | 393 |

[a]Spectral counts.

be a substantial pool of Ndc1 in the peripheral ER in certain cell types. Hypothetically, this pool could be engaged in functions other than NPC organization. In yeast, for example, Ndc1 has functions at the spindle pole body and at the NPC (Knockenhauer & Schwartz, 2016; Lin & Hoelz, 2019). In a related situation, a peripheral ER pool has been reported for emerin in certain cell types and conditions, and functions for emerin outside the INM have been described (Berk et al, 2013; Le et al, 2016).

A substantial number of the ER marker proteins had relatively high NE–ES1 values (~0.7–0.8), particularly proteins reported to be enriched in sheet ER (Ckap4, Rrbp1, Ssr1/4, Ktn1) (Shibata et al, 2010; Obara et al, 2022). However, the NE–ES2 values for these proteins were markedly lower, in most cases <0.5, because of the inclusion of HCM in the scoring calculation. These comparisons suggest that uncharacterized proteins with both NE–ES1 and NE–ES2 values >0.8 are strong candidates for consideration as NE-concentrated species. The reliability of these metrics increases with the level of spectral detection for individual proteins, with more confident assessments involving detection across many or all of the fractionation experiments (Table S1). NE–ES1 incorporates more spectral data in this regard, although it is less stringent than NE–ES2.

It is important to note that in most cases, NE-ES are meaningful only for proteins containing a TM domain(s). This is because many intranuclear proteins that are undetectable in isolated cytoplasmic membranes, including components of chromatin and ribonucleo-protein particles, co-fractionate with NEs to a minor extent and are not quantitatively extracted with urea or carbonate treatment. Because of their absence from cytoplasmic membranes, these contaminants have high NE-ES even though they are not concentrated at the NE in situ. Many proteins in this category were revealed by a previous analysis that included a "nuclear contents" fraction (Fig 1A) (Cheng et al, 2019), and usually are apparent from their annotations in Uniprot. In unusual cases, extraction-resistant behavior of membrane proteins might be dictated by protein features other than a TM domain, such as membrane-binding motifs, covalently attached lipids, and/or supra-molecular associations (Hurley, 2006). However, these features by themselves are not predictive, because efficient membrane extraction was seen for lamin B1, which is stably modified by farnesylation (Burke & Stewart, 2013), and for the nucleoporins Nup160, Nup133, Nup155, and Nup35, which contain membrane-associating amphipathic helices (Hamed & Antonin, 2021) (Table S2).

## Validation of candidate NE-concentrated proteins by quantitative immunofluorescence microscopy

We selected 16 TM proteins identified from the proteomics screen, most with NE–ES1 and/or NE–ES2 scores >0.9, to further evaluate as potential NE-concentrated proteins. For this, we used a cell-based targeting assay involving immunofluorescence localization (Table 1 and Figs 3 and S2). 14 of these (all except Tmem209 and Tmem214) were relatively low abundance based on their NSAF values, which

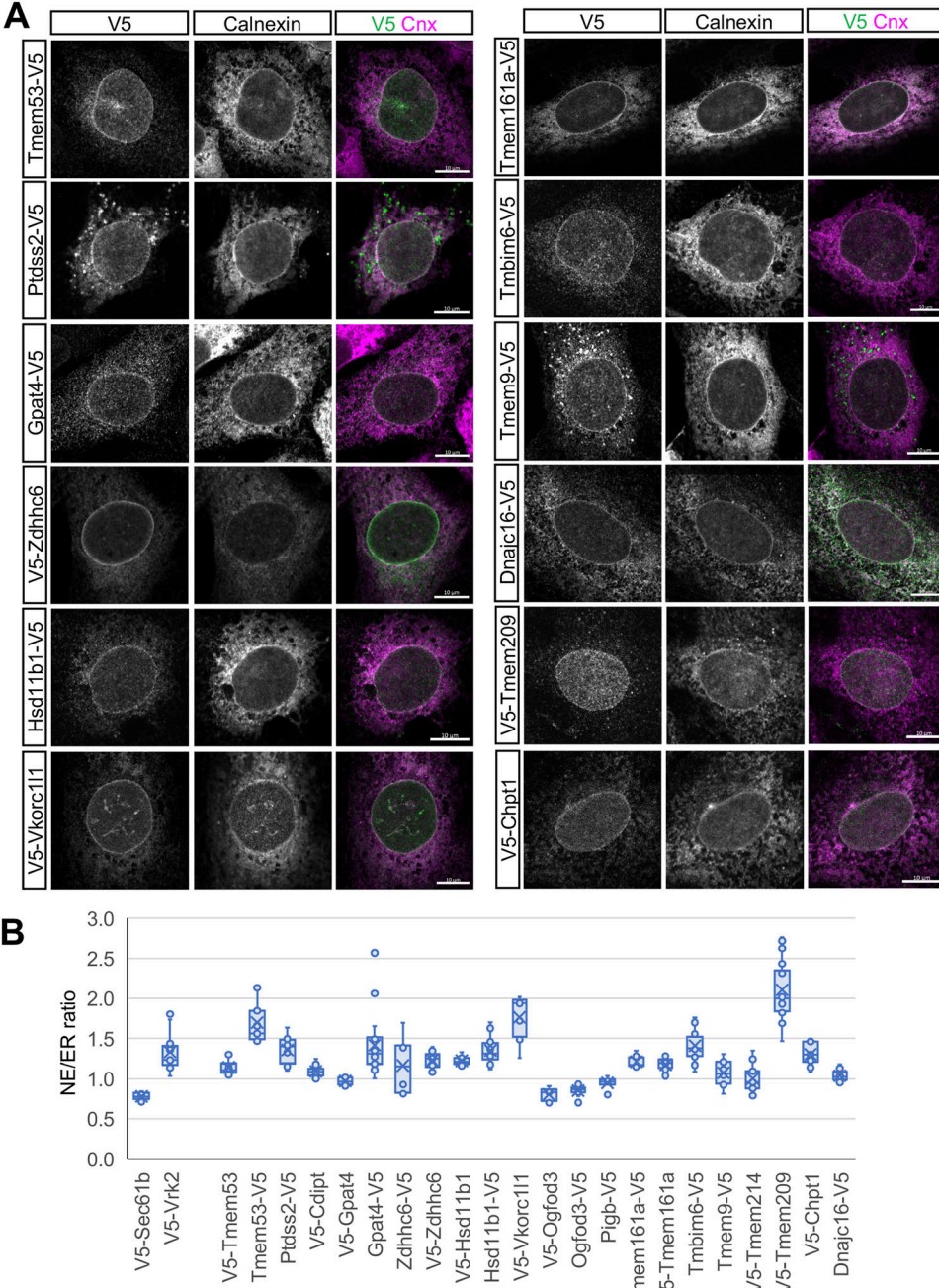

**Figure 3. Immunofluorescence localization of candidate NE-enriched proteins using an ectopic targeting assay.**

Proteins listed in Table 1 were tagged with a V5 epitope and transiently expressed in C3H cells using a lentiviral vector (see Fig S2). **(A)** Montage of immunofluorescence images of representative cells expressing the indicated constructs, co-stained with antibodies that recognize the V5 epitope and endogenous calnexin. A merged image is shown in the third panel of each set. Bars, 10 μm. **(B)** Box and whisker plot depicting NE/ER concentration ratio for each construct, as determined by quantification of immunofluorescence images using the method depicted in Fig S3. The NE/ER concentration of the control ER protein, V5-Sec61b, and of the previously identified NE-concentrated protein Vrk2 were determined in parallel. Data information: in (B), in most cases, at least nine cells were analyzed for each condition. Source data are available for this figure.

were at least fivefold–10-fold lower than the values for the TM nucleoporin markers Pom121 and Ndc1 (Fig S1 and Table S1). Two members of this group were reported to be NE-concentrated in certain cell types, Tmem209 (Schirmer et al, 2003; Malik et al, 2010; Fujitomo et al, 2012) and Tmem53 (Schirmer et al, 2003; Malik et al, 2010; Korfali et al, 2011), but the remainder have not been analyzed for NE concentration. The proteins had a range of sizes and predicted TM segments (Fig S2). Categorized by annotated functions, they included enzymes for lipid biosynthesis and modification, oxidoreductases, signaling regulators, and components involved in protein-folding and related activities (Table 1).

We used a lentiviral vector and transient transduction conditions (Tapia & Gerace, 2016) to achieve low-level expression of the epitope-tagged candidates in C3H cells, and deployed immunofluorescence microscopy to compare the subcellular localization of ectopic V5-tagged constructs with endogenous calnexin, an abundant TM marker that is uniformly distributed throughout the NE/ER system (Shibata et al, 2010) (Fig 3, see below). Low-level expression can be important for providing reliable NE-targeting results, because some proteins that are endogenously concentrated at the NE become uniformly distributed throughout the NE/ER network or appear in large cytoplasmic inclusions with higher

ectopic expression (Cheng et al, 2019). These effects, in part, likely reflect saturation of a limited number of binding sites for the proteins at the NE (Katta et al, 2014; Pawar & Kutay, 2021) and "spillover" of excess protein to contiguous ER elements. Whereas this ectopic targeting assay provides an accessible proxy for reporting the distribution of the endogenous proteins, the results reflect a composite of the rates of synthesis (in the peripheral ER) and NE targeting and turnover (in the two compartments), which might differ somewhat between endogenous proteins and ectopic counterparts.

For all 16 candidates analyzed, the ectopic constructs were localized to both the NE and peripheral ER, in a pattern broadly similar to calnexin (Figs 3A and S4). Although none of the constructs appeared in large aggregates with our conditions, Tmem9 and Ptdss2 did occur in smaller, heterogeneous cytoplasmic foci, and in ER-like structures. Visual inspection of the immunofluorescence patterns (Fig 3A) suggested that most of the candidates had a higher concentration at the NE than in the peripheral ER relative to calnexin. To analyze this quantitatively across populations of cells, we used a high information content method to calculate the relative concentration of ectopic constructs at the NE compared with the peripheral ER, designated the "NE/ER ratio" (Fig S3A). This method measured the fluorescence intensity of V5-tagged constructs (normalized to endogenous calnexin) in two circumferential bands around the nuclear periphery, one containing the NE and a second, more external band containing peripheral ER elements (Fig S3B and C).

From this quantitative analysis, most of the constructs (10 of the 16 proteins) had average NE/ER concentration ratios of at least 1.2 (Fig 3B), indicating that these proteins indeed were concentrated at the NE with our expression conditions. For some proteins, a substantially higher NE/ER ratio was observed with the epitope tag on the N- versus C-terminus (notably, Tmem53 and Gpat4). This may be explained by diminished epitope accessibility for the NE pool of the ectopic protein, or by tag-related inhibition of accumulation at the NE. Conversely, epitope tag placement had no effect on the NE/ER ratio for other proteins examined (Hsd11b1, Zdhhc6, Tmem161a). As a positive control, our quantification method yielded a NE/ER ratio of ~1.3 for Vrk2, a previously described NE-concentrated protein (Birendra et al, 2017; Cheng et al, 2019). By contrast, the peripheral ER marker Sec61β appeared to be less concentrated at the NE than the adjacent ER (NE/ER, ~0.8) as observed previously (Cheng et al, 2019), as were some of the candidates tested (Ogfod3, Pigb). We compared the NE targeting of Zdhhc6 and Hsd11b1 seen in transiently transduced C3H cells (Figs 3 and S3A) with that found in stably transduced C3H cells and MEFs (Fig S4B and C). We obtained similar targeting results with the different conditions, which all yielded NE/ER ratios of ~1.2–1.4.

The NE/ER ratio calculated by the above method generally underestimates the actual in situ NE concentration of the ectopic proteins. This is because the circumferential band used to define the NE reflects the average intensity of all structures in this region. Limited by the resolution of light microscopy, this zone likely contains peripheral ER elements juxtaposed to the ONM and/or parts of the tubular elements connecting these two structures (Obara et al, 2022) and the NE itself. Moreover, because all well-characterized NE proteins are concentrated at specific substructural regions of the NE (Schirmer et al, 2003; Cheng et al, 2019; Pawar & Kutay, 2021), the average NE concentration of specific proteins reported by this method would underrepresent their concentration at discrete NE subdomains (e.g., INM or NPC).

Determination of the localization of the proteins described here with regard to the NE substructure will require further analysis. However, the finely punctate labeling pattern on the nuclear surface seen for V5-Tmem209 was conspicuously reminiscent of NPC labeling (Fig 3, upper panel, Fig S5). Indeed, we observed strong co-localization of ectopic Tmem209 with the NPC-specific RL1 monoclonal antibody (Snow et al, 1987) (Fig S5), suggesting that Tmem209 may be a previously unrecognized TM nucleoporin (see the Discussion section).

In summary, the results from immunofluorescence localization of ectopically expressed constructs indicate that the majority of the candidates tested preferentially target to the NE relative to the peripheral ER with our conditions (see the Discussion section). Together with the proteomics data of membrane fractions, the results indicate that these proteins represent previously unrecognized NE-concentrated proteins.

### Functional analysis of the palmitoyltransferase Zdhhc6

An underlying prediction of this study is that some functions of NE-concentrated proteins are likely to involve the nucleus. To address this question, we carried out an initial analysis of the palmitoyltransferase Zdhhc6, one of the low-abundance NE proteins identified here. Out of the seven palmitoyltransferases detected with more than 30 spectral counts in our analysis, Zdhhc6 was the only one with a high NE-ES. We determined that a pool of Zdhhc6 is localized to the INM in the vicinity of lamin B1 using the proximity ligation assay (Fig S6), indicating that it potentially could function at the INM. Even though we did not detect Zdhhc6 in LCM or HCM fractions by proteomics, a substantial peripheral ER pool of this protein seems likely (see the Discussion section). This is because Zdhhc6 has been implicated in palmitoylation of multiple peripheral ER resident proteins including calnexin (Lakkaraju et al, 2012), the IP3 receptor (Fredericks et al, 2014), and the ubiquitin E3 ligase gp78 (Fairbank et al, 2012) (see the Discussion section).

To identify potential NE palmitoylation targets of Zdhhc6, we depleted Zdhhc6 from C3H cells with shRNA and metabolically labeled the cells with the palmitic acid analog 17ODYA. (Figs 4A and S7A). After cell lysis, 17ODYA-containing proteins were tagged with biotin using click chemistry (Gao & Hannoush, 2018), modified proteins were captured on streptavidin beads, and samples were analyzed by tandem mass tag (TMT) proteomics to quantitatively compare protein bound to streptavidin beads with the different conditions (Table S4). Palmitoylated proteins were revealed by their selective or increased detection in samples incubated with 17ODYA as compared with palmitic acid. Moreover, palmitoylated proteins that were dependent on Zdhhc6 were reduced in the Zdhhc6 knockdown relative to control cells. This analysis detected five INM proteins whose capture levels were consistently reduced with Zdhhc6 knockdown (Fig 4B and Table S4). We directly confirmed the palmitoylation of two of these proteins (lamin A and Tmx4), using Western blot analysis to detect biotinylated, 17ODYA-labeled proteins (Fig 4C). Lamin C, an alternative splice form of the gene for

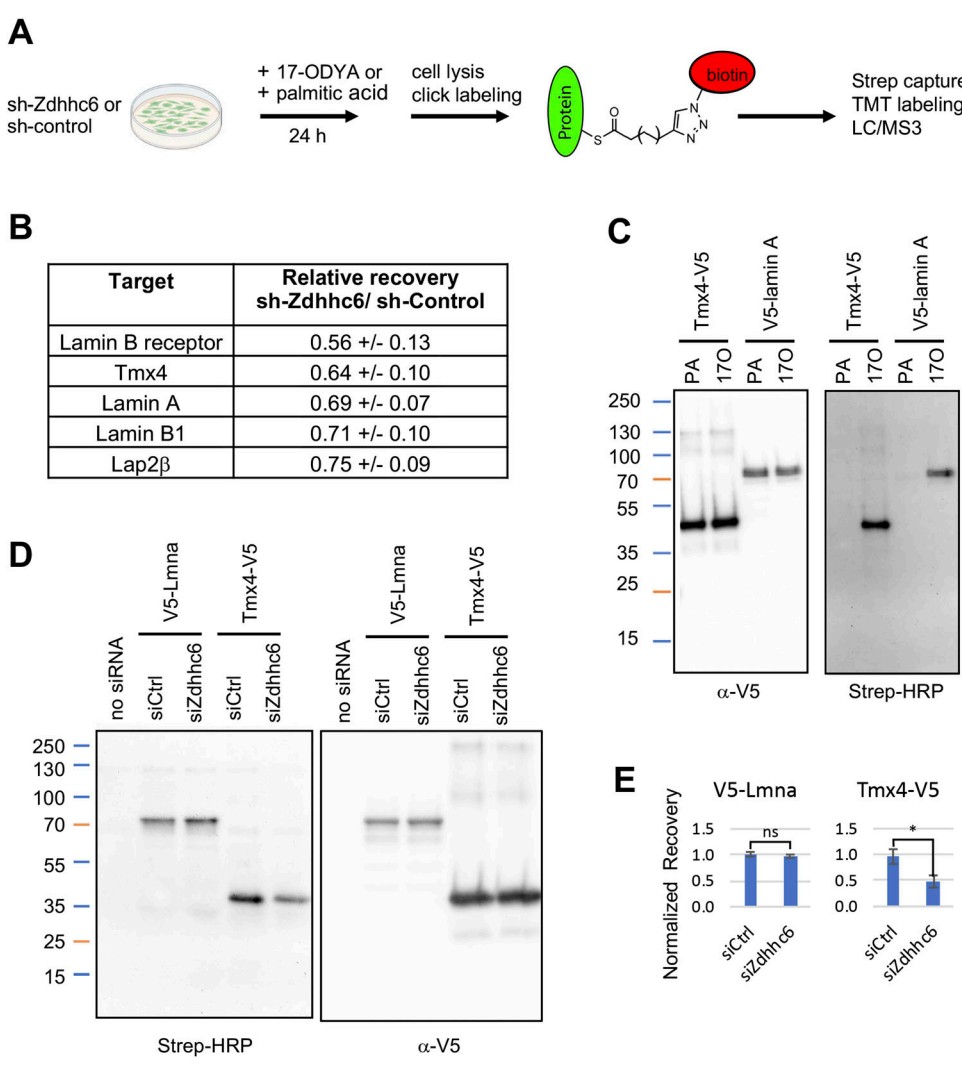

**Figure 4. Identification of proteins palmitoylated by Zdhhc6 using metabolic labeling.**
**(A)** Strategy used to identify proteins whose palmitoylation is dependent on Zdhhc6. C3H cells were stably transduced with an shRNA vector targeting Zdhhc6, or with either of two separate control shRNA vectors. Cells were incubated with either 17ODYA or palmitic acid (PA) for 24 h, lysed, and labeled by click chemistry to attach biotin to the alkyne group of 17ODYA. After complete solubilization with SDS, samples were incubated with streptavidin beads, which were washed and processed for on-bead digestion with trypsin. Eluted peptides were then labeled with TMT11 reagents and analyzed by LC/MS3. **(B)** The list of NE proteins detected in four experimental repeats that showed a substantial reduction in the level of bound protein with knockdown of Zdhhc6, indicating palmitoylation involving Zdhhc6. **(C)** Confirmation of Tmx4 and lamin A palmitoylation. **(A)** C3H cells that were stably transduced with vectors expressing Tmx4-V5 or V5-lamin A were incubated with 17ODYA (17A) or PA for 24 h, processed for click labeling with a biotin tag as in (A) and analyzed by SDS–PAGE followed by blot detection using streptavidin-HRP/α-HRP or α-V5 antibodies as indicated. **(D)** Palmitoylation of Tmx4 and lamin A with Zdhhc6 depletion. C3H cells stably expressing Tmx4-V5 or V5-lamin A were treated with siRNA-targeting Zdhhc6 or control siRNA. **(A, C)** Cells then were incubated with 17ODYA or PA for 24 h, and processed with click labeling and analysis as in (A, C). **(E)** Quantification of changes in palmitoylation of Tmx4 and lamin A with knockdown of Zdhhc6 by siRNA. Data information: in (B), values were derived from two independent cell/TMT11 samples that were analyzed in duplicate MS runs (four data points total). Data are presented as mean ± SEM. In (E), data are based on two repeats of the experiment shown in (D) and is presented as mean ± SEM. *P < 0.05 (t test); ns, not significant.

lamin A (Burke & Stewart, 2013), was not detectably palmitoylated in our analysis. Furthermore, we found that the modification level of Tmx4 was reduced by over 50% in cells with knockdown of Zdhhc6 by siRNA (Figs 4D and E and S7B). Together, these results indicate that Tmx4 is a palmitoylation target of Zdhhc6.

No significant changes in 17ODYA labeling of lamin A were seen with our conditions, suggesting that lamin A palmitoylation was not limited by the reduced level of Zdhhc6 obtained with transient knockdown by siRNA, in contrast to the effects obtained with stable depletion by shRNA (Fig S7 and Table S4). Previous screens have suggested that lamin A is palmitoylated (SwissPalm database), and the PalmPred program predicts palmitoylation of human lamin A on Cys588 and/or Cys592. Because these residues are conserved in mammals and in zebrafish, the functions of this modification merit further study.

Protein palmitoylation has widely reported effects on protein localization and stability (Linder & Deschenes, 2007; Lakkaraju et al, 2012; Lynes et al, 2012). To evaluate whether palmitoylation similarly influences Tmx4, we first mapped Tmx4 palmitoylation sites.

Calnexin, which is a well-characterized target of Zdhhc6, is modified on two cysteine residues positioned on the cytoplasmic side of the ER membrane, immediately adjacent to its TM segment (Lakkaraju et al, 2012; Dallavilla et al, 2016). Mouse Tmx4 contains two similarly positioned cysteine residues on residues 209 and 211, although only the Cys209 equivalent is conserved in human TMX4 (Fig 5A). We made mutated versions of mouse Tmx4 in which either one or both of the conserved cysteine residues were changed to alanine to block potential palmitoylation. Analysis of these constructs in cells labeled with 17ODYA revealed that most of the modification was lost with the C209A mutant, and that modification was virtually undetectable with the C209A/C211A mutant (Fig 5B). This suggests that the conserved C209 on Tmx4 is the main site of palmitoylation and that C211 is modified to a lower extent.

Regarding the functions of Tmx4 palmitoylation, we found that levels of the C209A, C211A, and C209A/C211A palmitoylation-deficient mutants were significantly increased in single-integration cell populations as compared with the WT protein (Fig 5C). Moreover, the NE/ER concentration ratio was significantly increased for the

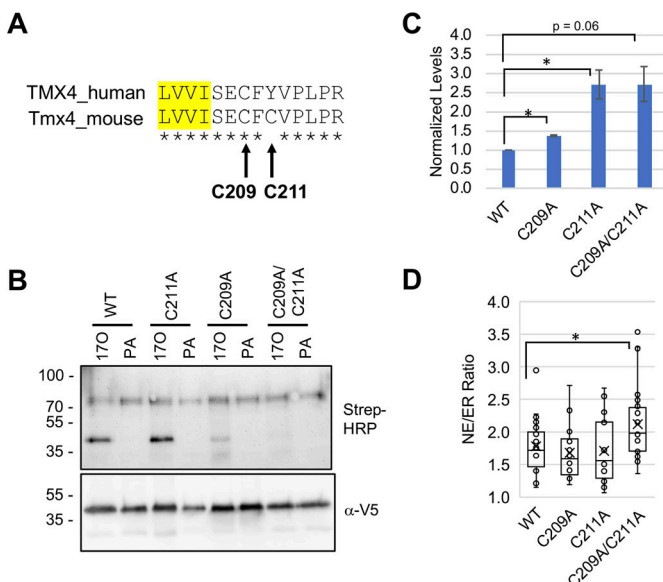

**Figure 5. Sites of Tmx4 palmitoylation and effects on protein levels and localization.**
**(A)** Sequence of human and mouse Tmx4 near potential palmitoylation sites. Yellow shading indicates the C-terminal end of the transmembrane domain, with the cytoplasmically oriented segment to the right. **(B)** Palmitoylation in C3H cells stably transduced with V5-tagged versions of WT Tmx4 or with the Tmx4 point mutants indicated. Cells were labeled for 24 h with either 17ODYA (17O) or palmitic acid, processed by click labeling with a biotin tag and isolation on streptavidin–sepharose, and analyzed by SDS–PAGE with blot detection using streptavidin-HRP/α-HRP or α-V5 antibodies as indicated. Results similar to those shown were obtained in three experiments. **(C)** Quantification of the level of V5-tagged Tmx4 constructs in stably transduced C3H populations with single-copy lentiviral integrations of the various constructs, using SDS–PAGE and Western blotting with α-V5 antibodies. **(D)** Box and whisker plot depicting the NE/ER concentration of various Tmx4 constructs in stably transduced C3H cell populations with single-copy lentiviral integration. Data information: in (C), data are presented as mean ± SEM. *P < 0.05 (t test). In (D), at least 10 cells were analyzed for each condition.

C209A/C211A mutant as compared with WT Tmx4 (Fig 5D). Together, these results suggest that palmitoylation both reduces the cellular levels of Tmx4 and diminishes its relative concentration at the NE. This points to palmitoylation as a means of controlling the levels of Tmx4 at the INM by two complementary mechanisms, and strongly suggests that Zdhhc6 has nucleus-related functions at the NE.

## Discussion

To promote an understanding of NE functions, we have refined methods to identify low-abundance TM proteins that are concentrated at the NE. The core of our approach involved the use of label-free proteomics to analyze membrane fractions that were chemically extracted to enrich for TM proteins. By comparing isolated NEs with membrane fractions containing cytoplasmic organelles that partially co-fractionate with NEs, we identified proteins with selectively high NE abundance as defined quantitatively by enrichment scores. To validate these candidates, proteins with high NE-ES were ectopically expressed at low levels in

cultured cells to quantify targeting to the NE from their site of synthesis in the peripheral ER using immunofluorescence microscopy.

The proteomics screen revealed numerous candidates with high NE-ES (>0.8) that were not previously connected to NE functions. We selected a diverse sample set of candidates for validation, most of which were low abundance. These included two proteins previously reported to be concentrated at the NE in certain cell types, Tmem209 (Schirmer et al, 2003; Malik et al, 2010; Fujitomo et al, 2012) and Tmem53 (Schirmer et al, 2003; Malik et al, 2010; Korfali et al, 2011). Both of these proteins targeted strongly to the NE in C3H cells as shown by our quantitative analysis. They have orthologs in a diverse range of animals including flies and worms (PANTHER resource). TMEM53 has been linked to a bone disorder in humans that is very similar to a disease caused by mutations in the INM protein MAN1 (LEMD3) (Guo et al, 2021), consistent with an important function at the NE. Interestingly, we found that ectopically expressed Tmem209 yielded punctate immunofluorescence labeling at the NE that strongly co-localized with NPCs. It was the only protein examined in the ectopic targeting assay with this characteristic. Moreover, Tmem209 has a plant ortholog that is suggested to be NPC-associated (Goto et al, 2019; Tang et al, 2020). Because its abundance in C3H cells is similar to that of the TM nucleoporins Pom121 and Ndc1 (Table S1), Tmem209 may be a previously unrecognized architectural component of the NPC in animals that merits further analysis.

Regarding the 14 other candidates that we analyzed with ectopic expression, we found that eight of these targeted preferentially to the NE by immunofluorescence microscopy. Extrapolating from these results, it is likely that a substantial number of additional low-abundance proteins with high NE-ES also are enriched at the NE in situ. A priori, most of the NE-concentrated proteins we identified are likely to be concentrated at the INM and/or nuclear pore membrane, rather than the ONM (see the Introduction section).

Six of the candidates with high NE-ES did not substantially concentrate at the NE in the ectopic targeting assay. It is possible that (some of) these were false positives that received high scores because of disproportionally low detection in the LCM and HCM fractions by MudPIT proteomics. Alternatively, they might indeed be NE-concentrated in C3H cells, but eluded detection because of unavoidable limitations of the ectopic targeting assay. First, the epitope tag might impair protein detection and/or targeting to the NE. This is exemplified by the considerably stronger NE concentration measured for the C-terminally tagged versions of Tmem53 and Gpat4 than the N-terminal versions. Second, ectopically overexpressed proteins might be undetectable above the levels that populate the peripheral ER if their NE accumulation is specified by a relatively low number and/or affinity of binding sites. Finally, the accumulation of certain proteins at the NE might require coordinated expression with other NE-binding partners (e.g., other members of a multimeric complex) that is not obtained with the ectopic conditions. Interestingly, two proteins with high NE-ES that did not show NE concentration with our expression conditions, Tmem9 and Tmem214, were reported to interact with multiple NE proteins in stringent affinity capture-mass spectrometry screens involving cultured cells (Hein et al, 2015; Huttlin et al, 2021). This is consistent with a NE association for these proteins that was not detected by our methods.

Whereas our proteomics and ectopic targeting methods identified proteins that are relatively concentrated at the NE versus the peripheral ER, a precise determination of their copy numbers in the two compartments in situ remains a formidable challenge. Genomic tagging of proteins (Cho et al, 2022) or the use of specific antibodies in principle could address this issue, but both methods are currently limited by sensitivity for detection of low-abundance proteins. Because the NE contains less than 5–10% of the surface area of the peripheral ER in most mammalian cells (Shibata et al, 2010; Obara et al, 2022), many NE-concentrated proteins may have a *greater copy number* in the peripheral ER than in the NE, despite their *selective concentration* at the NE. Viewed from this perspective, many or all of the NE-enriched proteins may have functions in the peripheral ER and at the NE (see below).

The protein set that we validated with the NE-targeting assay spans multiple functional categories, including lipid biosynthesis, redox regulation, and signaling control. Whereas the peripheral ER is a well-established site for lipid biosynthesis (Jacquemyn et al, 2017), the INM can be an important site for promoting synthesis of phosphatidylcholine and triglycerides under some conditions (Ohsaki et al, 2016; Romanauska & Kohler, 2018). Our findings that certain phosphoglyceride synthesis isozymes are concentrated at the NE (Chpt1, Ptdss2, Gpat4) expand this current insight. Enrichment of Chpt1 at the INM is consistent with the finding that Pcy1a, the enzyme that synthesizes CDP-choline and is rate-limiting for phosphatidylcholine synthesis, translocates from the nucleoplasm to the NE under conditions of phosphatidylcholine deficit (Cornell & Ridgway, 2015; Haider et al, 2018). Conceivably, the relative NE concentration of Chpt1 increases under similar conditions. Moreover, our observed NE concentration of Gpat4, the enzyme that catalyzes attachment of the first fatty acyl chain to glycerolphosphate (Jacquemyn et al, 2017), suggests that other downstream steps in phosphoglyceride biosynthesis may partly occur at the INM, in addition to the reactions catalyzed by Chpt1 and Ptdss2. Although the functional significance of lipid biosynthesis at the INM remains speculative, it can afford a mechanism to enhance coordination between the transcription of lipid biosynthetic genes and the end products of their action (Jacquemyn et al, 2017).

Our detection of signaling regulators concentrated at the NE, including Tmbim6 and Tmem53, suggests a broader role for the NE in regulation of signaling than previously appreciated (Gerace & Tapia, 2018). Tmbim6 is a Bax interactor that inhibits apoptosis in several cultured cell models (Lebeaupin et al, 2020), and Tmem53 regulates cell cycle progression via several pathways (Korfali et al, 2011) including a recently described role in TGFβ-Smad signaling (Guo et al, 2021). Unraveling the mechanistic basis for these effects now can be considered in the context of the NE environment.

We previously found that the oxidoreductase Tmx4 is concentrated at the INM (Cheng et al, 2019), where it potentially could regulate the LINC complex and/or associated TorsinA, redox-sensitive proteins involved in connecting the nucleus to the cytoplasmic cytoskeleton (Lu et al, 2008; Zhu et al, 2010; Cain et al, 2018). Indeed, the oxidoreductase function of Tmx4 was recently implicated in disassembly of the LINC complex under pharmacological stress to induce NE autophagy (Kucinska et al, 2022 *Preprint*). Our identification of Vkorc1l1 as an additional NE-concentrated oxidoreductase may expand this functional network because Vkorc1, a protein with a high NE-ES that is closely related to Vkorc1l1, interacts in a redox reaction with Tmx4 (Schulman et al, 2010). In a separate functional context, the NE enrichment of Hsd11b1, which reductively activates glucocorticoids and other steroids (Swarbrick et al, 2021), suggests a potential nuclear control system for positive regulation of steroid hormone action. Such a system also might include Steryl-sulfatase (Sts), another protein from our screen with a high NE-ES that was not examined in the targeting assay.

We focused on Zdhhc6 to investigate whether a low-abundance protein concentrated at the NE could have nucleus-targeted functions. Our data indicated that several INM proteins including lamin A and the Tmx4 are its palmitoylation targets. Regarding Tmx4, we identified the major sites of Tmx4 palmitoylation by Zdhhc6 by mutagenesis and found that palmitoylation-deficient mutants were significantly increased in both steady-state levels and in their NE/ER concentration ratio. Accordingly, palmitoylation of Tmx4 may attenuate Zdhhc6 functions at the INM both by decreasing its overall levels (most likely by increased proteosomal turnover [Dallavilla et al, 2016]) and by decreasing its INM concentration relative to the peripheral ER. This work strongly suggests a NE-selective function for Zdhhc6. Zdhhc6 itself is regulated by palmitoylation (Abrami et al, 2017), underscoring the potential regulatory complexity of this system.

In conclusion, the approaches we have outlined here, utilizing proteomic analysis of isolated membrane fractions and quantitative ectopic targeting assays, provide a template for confident identification of NE-concentrated proteins. Our findings have identified new NE-concentrated proteins that represent multiple functional categories, and have revealed a substantial number of additional candidates that remain to be directly evaluated. This information can provide a valuable framework for generating and testing new hypotheses on NE functions that heretofore have not been addressed.

## Materials and Methods

### Cell culture

C3H/10T1/2 (C3H) cells (#CCL-225; ATCC) and 293T cells (#CRL-3216; ATCC) were acquired from the American Type Culture Collection and used at low passage number after freezing expanded stocks. MEFs were generated from C57BL/6 mice by immortalization with the SV40 T antigen. MEFs, C3H, and 293T cells were maintained in high-glucose DMEM (Gibco) supplemented with 10% FBS (Gibco), 1% penicillin/streptomycin/glutamine cocktail (Gibco), and 1% minimum essential medium nonessential amino acids (NEAA) (Gibco). All cells were maintained at 37°C in 5% $CO_2$.

### Subcellular fractionation and membrane extraction

For subcellular fractionation, C3H cells were seeded in 500 $cm^2$ plates and allowed to reach 90% confluency. Plates were rinsed three times with ice-cold PBS, and then three times with ice-cold homogenization buffer (HB) (10 mM HEPES pH 7.8, 10 mM KCl, 1.5 mM $MgCl_2$, 0.1 mM EGTA, 1 mM DTT, 1 mM PMSF, and 1 μg/ml each of

pepstatin, leupeptin, and chymostatin). After these washes, cells were incubated in HB for 15 min on ice. Cells then were scraped off plates and were further disrupted by dounce homogenization with 18–20 strokes to achieve >95% cell disruption. The whole cell homogenate was layered on top of a 2-ml shelf of 0.8 M sucrose in HB and centrifuged at 1,000$g$ for 10 min at 4°C in a JS5.2 swinging bucket rotor (Beckman Coulter) to yield a low-speed nuclear pellet and a postnuclear supernatant, the latter comprising the zone above the sucrose shelf. The low-speed nuclear pellet was resuspended in 1.8 M sucrose in HB using a cannulus, and the postnuclear supernatant was adjusted to a final concentration of 1.8 M sucrose in HB. The resuspended low-speed nuclear pellet and the postnuclear supernatant were each layered in separate ultra-clear 13.2 ml nitrocellulose centrifuge tubes on top of a 1-ml layer of 2.0 M sucrose in HB. For the nucleus gradient, HB was layered over the loading zone to fill the nitrocellulose tube. For the postnuclear supernatant gradient, 1 ml of 1.4 M sucrose in HB was layered on top of the loading zone, followed by HB to fill the tube. The gradients then were centrifuged at 210,000$g$ for 1 h at 4°C with no brake in a SW41Ti swinging bucket rotor (Beckman Coulter). Two samples were collected from the nucleus gradient: a fraction comprising the HB/1.8 M sucrose interface, termed "heavy cytoplasmic membranes 2" (HCM2), and the pellet at the bottom of the 2.0 M sucrose layer, which comprised the purified nuclei fraction. The latter was collected by resuspension in HB and dounce homogenization with two strokes to disperse aggregates. For the postnuclear supernatant gradient, the HB/1.4 M sucrose interphase was collected and saved as LCM and the 1.4 M sucrose/1.8 M sucrose interface was collected and saved as HCM. To prepare NEs, resuspended purified nuclei were incubated with 1 mM CaCl$_2$ and 100 ku/ml micrococcal nuclease (New England Biolabs) in HB for 37°C for 15 min. Digested nuclei were then placed on ice and NaCl was added to a final concentration of 500 mM. The sample then was layered on top of a 1-ml shelf of 0.8 M sucrose in HB and centrifuged at 4,000$g$ for 10 min at 4°C in a JS5.2 rotor. A sample comprising the region above the 0.8 M sucrose layer was collected and saved as "nuclear contents." The pellet was collected with resuspension in HB and saved as the NE fraction.

Chemical extraction of isolated membrane fractions involved four separate cell/membrane preparations. Isolated membrane fractions were dounce homogenized briefly and the protein concentration of each fraction was adjusted 0.5 mg/ml based on the Pierce BCA assay (Thermo Fisher Scientific). To carry out the chemical extractions, 200 $\mu$l of each membrane fraction was added to 1.8 ml of the following solutions: deionized water (pre-wash), 100 mM sodium carbonate pH 11.5 (Cw) or 8 M urea in 25 mM NaCl, 20 mM Tris–HCl pH 8.8 (Uw). For the Cw, samples were incubated for 30 min at 4°C. For the Uw, samples were incubated 30 min at room temperature. Samples were centrifuged for 1 h at 239,000$g$ at 4°C with a TLA100.3 fixed angle rotor. Pellets were gently rinsed with ice-cold deionized water and then processed for proteomics. Two membrane preparations were divided and used for parallel carbonate and urea extractions, and an additional two membrane preparations were used for urea extraction only.

## MudPIT proteomics and NE-enrichment scoring

MudPIT MS/MS proteomics (Wolters et al, 2001) was carried out on NE and LCM fractions obtained from all four of the membrane

preparations, and on HCM and HCM2 fractions obtained from three of these preparations. 30 $\mu$g of protein from each subcellular fraction, estimated by the BCA protein assay, was suspended in 4 M urea, 0.2% RapiGest SF (Waters Corporation), and 100 mM NH$_4$HCO$_3$ pH 8.0. Proteins were reduced with Tris(2-carboxyethyl)phosphine hydrochloride (TCEP) and alkylated with 2-chloroacetamide. Next, proteins were digested with 0.5 $\mu$g Lys-C (Wako) for 4 h at 37°C, and then for 12 h at 37°C in 2 M urea, 0.2% RapiGest SF, 100 mM NH$_4$HCO$_3$ pH 8.0, 1 mM CaCl$_2$ with 1 $\mu$g trypsin (Promega). Digested proteins were acidified with TFA to pH < 2 and RapiGest SF was precipitated out. Each fraction was loaded on individual MudPIT micro-columns (2.5 cm SCX: 5 $\mu$m diameter, 125 Å pores; and 2.5 cm C18 Aqua: 5 $\mu$m diameter, 125 Å pores; Phenomenex), and resolved across an analytical column (15 cm C18 Aqua: 5 $\mu$m diameter, 125 Å pores) (Phenomenex).

Analysis was performed using an Agilent 1200 HPLC pump and a Thermo LTQ-Orbitrap Velos Pro using an in-house built electrospray stage. MudPIT experiments were performed with steps of 0%, 10%, 20%, 30%, 50%, 70%, 80%, 90%, 100% buffer C and 90/10% buffer C/B (Wolters et al, 2001), being run for 5 min at the beginning of each gradient of buffer B. Electrospray was performed directly from the analytical column by applying the ESI voltage at a tee (150 mm ID) (Upchurch Scientific) (Wolters et al, 2001). Electrospray directly from the LC column was done at 2.5 kV with an inlet capillary temperature of 325°C. Data-dependent acquisition of tandem mass spectra was performed with the following settings: MS/MS on the 20 most intense ions per precursor scan; 1 microscan; reject unassigned charge state and charge state 1; dynamic exclusion repeat count, 1; repeat duration, 30 s; exclusion list size 500; and exclusion duration, 90 s.

Protein and peptide identification was done with the Integrated Proteomics Pipeline—IP2 (Bruker Corp. https://bruker.com/). Tandem mass spectra were extracted (mono-isotopic peaks) from raw files using RawConverter (He et al, 2015) and were searched against the UniProt SwissProt *Mus musculus* database (release 2018_01) with reversed sequences using ProLuCID (Peng et al, 2003; Xu et al, 2015). The search space included all fully tryptic and half-tryptic peptide candidates with static modification of 57.02146 on cysteines. Peptide candidates were filtered using DTASelect (Cociorva et al, 2007) at 1% protein level false discovery rate, (parameters: -p 1 -y 1 --trypstat --pfp 0.01 --extra --pI -DM 10 --DB --dm -in -t 0 --brief –quiet) (Tabb et al, 2002; McDonald et al, 2004).

The NSAF was used to estimate the relative abundance of individual proteins detected by MudPIT MS/MS (Zybailov et al, 2005). To determine NSAF values for each MudPIT run, the spectral count per unit (amino acid) length of all proteins detected was calculated, and this value for each protein was normalized to the summation of this value for all proteins identified in the same MudPIT run, thereby describing a normalized abundance of each protein. The NSAF values for each protein in the various membrane fractions analyzed were used to calculate NE-ES. NE enrichment score 1 used data from NE and LCM fractions obtained for all four membrane preparations and for both urea and carbonate extraction conditions. It involved the following equation, where e = each specific experimental run and p = protein ID:

$$NE\text{-}ES1_p = \frac{\sum_e NSAF_{NEep}}{\sum_e \left( NSAF_{NEep} + NSAF_{LCMep} \right)}.$$

NE enrichment score 2 (NE-ES2) used the additional data for HCM and HCM2 fractions that were obtained from three of these preparations according to the following:

$$NE\text{-}ES2_p = \frac{\sum_e NSAF_{NEep}}{\sum_e \left( NSAF_{NEep} + NSAF_{LCMep} + NSAF_{HCMep} + NSAF_{HCM2ep} \right)}.$$

### Isolation of palmitoylated proteins by metabolic labeling

C3H cell populations were stably transduced with a lentiviral vector that produced either (a) shRNA targeting Zdhhc6 ("shZdhhc6"), (b) a control shRNA sequence not found in the mouse genome ("shCtrl") or (c) a second control shRNA targeting GFP ("shGFP"). Cells were used for metabolic labeling within five passages after lentiviral transduction/puromycin selection. Two biological replicates of the three cell populations were each analyzed with two LC/MS3 technical repeats. For each replicate, 5 x $10^5$ cells per 10-cm dish were seeded 1 d before metabolic labeling. On the day of labeling, cells were rinsed with PBS and 10 ml of growth medium containing 50 $\mu$M 17-octadecynoic acid (17ODYA) or 50 $\mu$M phosphatidic acid was added. After 24 h, cells were harvested by trypsinization and pelleting, and were washed twice with cold PBS. The cells then were resuspended in non-denaturing lysis buffer (20 mM Tris–HCl pH 8, 300 mM NaCl, 2 mM EDTA, 1% NP-40, supplemented with 10 $\mu$M palmostatin B [#17851; Merck] and Halt phosphatase and protease inhibitor cocktails [#78440; Thermo Fisher Scientific]) and were sonicated on ice for two cycles of 10 s with a Sonic Dismembrator 60 (Thermo Fisher Scientific). The lysate was clarified by centrifugation at 4°C for 15 min at 20,000g.

The clarified lysate then was precipitated with methanol–chloroform (4:1) to remove free 17ODYA, and 1 mg of protein from each sample was click-labeled with Biotin-PEG3-azide (Cayman) (0.58 mM Biotin-PEG3-azide, 1.15 mM TCEP, 1.15 mM CuSO$_4$, 1X Tris((1-benzyl-4-triazolyl)methyl)amine solution in 1:4 DMSO/butanol) by incubating for 1 h at room temperature. Free biotin-PEG3-azide was removed by methanol–chloroform precipitation and the protein pellets were resuspended in 6 M urea supplemented with 2% SDS in PBS. Protein samples were reduced for 20 min with 5 mM TCEP and then were alkylated with 20 mM iodoacetamide in the dark. Samples then were diluted 10-fold with PBS and were incubated with 100 $\mu$l of a 50% streptavidin–sepharose (Cytiva) bead slurry for 1.5 h at room temperature, washed four times with 2 M urea in PBS containing 0.2% SDS pH 7, four times with 2 M urea in PBS pH 7, and finally washed once with 50 mM tetraethylammonium bromide. Protein-bound streptavidin–sepharose beads were then stored at –80°C until further processing.

### TMT sample preparation, data acquisition, and analysis

Frozen streptavidin–sepharose beads were resuspended in 100 mM tetraethylammonium bromide, 8 M urea, reduced with 5 mM TCEP, and alkylated with 10 mM 2-chloroacetamide. The bead suspension was diluted to 2 M urea, supplemented with 1 mM CaCl$_2$, and digested with 2 $\mu$g trypsin (Promega) for 15 h at 37°C. Beads were pelleted by centrifugation at 20,000g for 5 min, and 150 $\mu$g of protein was removed from the supernatant. After pooling 1/10$^{th}$ of each sample to create a reference, each sample was labeled with 11-plex TMT (Thermo Fisher Scientific) at 3:1 (TMT:protein) ratio in 30% acetonitrile, according to the manufacturer's recommendation. After labeling, the samples were pooled and acetonitrile was removed using vacuum centrifugation. The mixture of TMT-labeled peptides was acidified and fractionated using a high-pH reversed-phase peptide fractionation kit (Thermo Fisher Scientific) according to the manufacturer's recommendation. The fractions were dried by vacuum centrifugation and resuspended in 5% acetonitrile, 0.1% formic acid.

TMT-labeled peptides were analyzed using an EASY-nLC 1200 UPLC coupled with an Orbitrap Fusion Lumos mass spectrometer (Thermo Fisher Scientific). LC buffer A (0.1% formic acid, 5% acetonitrile in H$_2$O) and buffer B (0.1% formic acid, 80% acetonitrile in H$_2$O) were used for all analyses. Peptides were loaded on a C18 column packed with Waters BEH 1.7 $\mu$m beads (100 $\mu$m × 25 cm, tip diameter 5 $\mu$m), and separated across 180 min: 1–40% B over 140 min, 40–90% B over 30 min and 90% B for 10 min, using a flow rate of 400 nl/min. Eluted peptides were directly sprayed into MS via nESI at ionization voltage 2.8 kV and source temperature 275°C. Peptide spectra were acquired using the data-dependent acquisition synchronous precursor selection-MS3 method. Briefly, MS scans were done in the Orbitrap (120k resolution, automatic gain control AGC target 4 × $10^5$, max injection time 50 ms, m/z 400–1,500), the most intense precursor ions at charge states 2–7 were then isolated by the quadrupole and CID MS/MS spectra were acquired in the ion trap in Turbo scan mode (isolation width 1.6 Th, CID collision energy 35%, activation Q 0.25, AGC target 1 × $10^4$, maximum injection time 100 ms, dynamic exclusion duration 10 s), and finally, 10 notches of MS/MS ions were simultaneously isolated by the orbitrap for synchronous precursor selection HCD MS3 fragmentation and measured in the orbitrap (60k resolution, isolation width 2 Th, HCD collision energy 65%, m/z 120–500, maximum injection time 120 ms, AGC target 1 × $10^5$, activation Q 0.25).

Protein and peptide identification was done with the Integrated Proteomics Pipeline—IP2 (Bruker Corp. https://bruker.com/). Tandem mass spectra were extracted (monoisotopic peaks) from raw files using RawConverter (He et al, 2015) and were searched against a UniProt SwissProt *M. musculus* database (2020_01 release), including streptavidin (UniProt #P22629), with reversed sequences and standard contaminants, using ProLuCID (Peng et al, 2003; Xu et al, 2015).

The database used for the proteomics search can be found at: https://massive.ucsd.edu/ProteoSAFe/private-dataset.jsp?task=417b3a6d8c4e4809b247dfcbb728de32. The search space included all fully tryptic and half-tryptic peptide candidates with static modification of 57.02146 D on cysteines and 229.1629 D on lysines and N-termini. Peptide candidates were filtered using DTASelect (Cociorva et al, 2007) at 1% spectrum level false discovery rate, (parameters: -p 2 -y 1 --trypstat --fp 0.01 --extra --pI -DM 10 --DB --dm -in -t 1 --brief –quiet) (Tabb et al, 2002; McDonald et al, 2004).

## Molecular cloning

To construct lentiviral vectors expressing V5-tagged versions of our proteins of interest, the following cDNA clones were purchased from Origene: pCMV6-Chpt1 (RefSeq NM_144807), pCMV6-Pigx, (NM_024464)and pCMV6-Dnajc16 (NM_172338). The remaining genes were cloned from a cDNA library constructed from the C3H cells. Genes of interest were amplified by PCR using Q5 high-fidelity DNA polymerase (#M0491L; New England Biolabs). All genes were inserted into either pLV-Ef1a-V5-LIC-IRES-PURO (#120247; Addgene) or pLV-Ef1a-LIC-V5-IRES-PURO (#120248; Addgene). pLV-EF1a-IRES-Puro LIC-compatible vectors were digested with SrfI (New England Biolabs). PCR fragments and SrfI-digested vector were combined with NEBuilder HiFi DNA Assembly Master Mix (#E2621; New England Biolabs) in a 2:1 ratio of insert to vector. The DNA and DNA Assembly Master Mix were incubated at 50°C for 20 min and then NEB Stable Competent Cells (New England Biolabs) were transformed with the product. Cells were incubated at 30°C for 24 h, and the resulting colonies were picked for clone validation. All cDNA clones were confirmed by complete DNA sequencing of the ORF in both 5′-3′ and 3′-5 directions. All clones for lentiviral vectors expressing the proteins described in this study are available from Addgene.

## Lentiviral transduction of cells

To produce lentiviruses, 293T cells at 60–80% confluency were shifted to DMEM supplemented with 10% FBS, 1% glutamine, and 1% NEAA without antibiotics 30 min before transfection. Cells were transfected with pRSV-REV (gift from Didier Trono, #12253; Addgene plasmid), pMDL-RRE (gift from Didier Trono, #12251; Addgene plasmid), pCMV-VSVg (gift from Bob Weinberg, #8454; Addgene plasmid), and a pLV-EF1a-gene-of-interest vector (pLV-EF1a-GOI) using Lipofectamine 2000 (Invitrogen). Viral supernatant was harvested 48 h after transfection and filtered through a 0.45-$\mu$m polyethersulfone membrane filter (GE Healthcare Whatman). Western blotting of the viral supernatants with anti-V5 antibodies (below) was used to assess expression of the encoded V5-tagged constructs.

For stable lentiviral transduction of C3H cells and MEFs, cells were diluted to 5 x 10$^4$ cells/ml in DMEM with 10% FBS, 1% glutamine, and 1% NEAA without antibiotics, and polybrene (EMD Millipore) was added to a final concentration of 10 mg/ml. Cells were transduced with different viral loads (ranging from 1–500 $\mu$l of viral supernatant per 1 ml of cells) to obtain cell populations with a range of multiplicities of infection. After 3 d, cells were treated with puromycin (Invitrogen) to select for cells that had integrated viral DNA. C3H cells and MEFs were treated with 5 $\mu$g/ml and 1 $\mu$g/ml puromycin, respectively, for up to 1 wk. Cell populations with <30% cell survival under puromycin treatment, reflecting mostly single-integration events, were further expanded and grown for Western blotting and immunofluorescence. Transient lentiviral transduction of C3H cells involved the methods described above, except cells that were analyzed 48 h after treatment with virus without puromycin selection.

## Depletion of Zdhhc6 by RNAi

Analysis of the palmitoylation targets of Zdhhc6 by proteomics used C3H cell populations that were stably transduced with lentiviral vectors producing shRNA, using the methods described above.

Cells with depletion of Zdhhc6 were transduced with sh-Zdhhc6 (#RMM3981-201743749; Dharmacon). Control cells were transduced with sh-eGFP (#RHS4459; Dharmacon) and non-targeting control shRNA (#RHS6848; Dharmacon) (see Fig 4). Levels of Zdhhc6 mRNA in the stable populations were determined by qRT–PCR as described (Cheng et al, 2022).

Subsequent experiments with Zdhhc6 depletion involved the use of C3H populations that had been stably transduced with various V5-tagged Tmx4 lentiviral constructs. Zdhhc6 depletion in these cells was achieved with ON-TARGETplus siRNA SmartPool oligonucleotides, one targeting Zdhhc6 (#L-059510-01-0005; Horizon Discovery) and a second comprising a non-targeting control pool siRNA (#D-001810-10-05; Horizon Discovery). For this procedure, siRNAs were dissolved in 60 mM KCl, 6 mM HEPES pH 7.5, 0.2 mM MgCl$_2$ and added to serum-free DMEM to a concentration of 50 nM. This solution was then diluted to a final concentration of 25 nM siRNA with a 1:4 mixture of DharmaFECT-1 (DF-1) reagent (#T-2001-03; Horizon Discovery) and DMEM, respectively, and was incubated at room temperature in the dark for 20 min. Using C3H populations grown to a density of 5 × 10$^5$ cells per 10-cm plate, the siRNA-DF-1 mixture was added drop-wise to cells to a final concentration of 5 nM and distributed evenly with gentle rocking. After growth for 24 h, the medium was replaced with DMEM supplemented with serum and antibiotics. At 48 h posttransfection, cells were analyzed by Western blotting and qRT–PCR (Cheng et al, 2022).

## Antibodies

The following primary antibodies/reagents were used for Western blotting: rabbit anti-calnexin (#C4731; Sigma-Aldrich), rabbit anti-lamin A (affinity purified, made in-house to residues 396–429 of human lamins A and C) (Schirmer et al, 2001), rabbit anti-LAP2$\beta$ (made in-house to residues 1–194 of rat LAP2$\beta$), mouse anti-Tim23 (#611222; BD Transduction Laboratories), streptavidin-HRP (S911; Thermo Fisher Scientific), mouse anti-actin (clone C4, gift from Dr. Velia Fowler), mouse anti-myosin heavy chain (clone 3J14, #M9850-15B; US Biological), mouse anti-vimentin (clone RV202, #ab8978-1; Abcam). The secondary antibodies used for Western blot detection were: sheep anti-mouse HRP (#NA931; GE Healthcare), donkey anti-rabbit HRP (#NA934V; GE Healthcare).

The following primary antibodies were used for immunofluorescence staining: mouse anti-V5 (#46-0705; Invitrogen), rabbit anti-V5 (#PA1-993; Thermo Fisher Scientific), rabbit anti-calnexin (#ab22595; Abcam), guinea pig anti-lamin A/C (made in-house to gel-purified rat liver lamin A), mouse monoclonal IgM RL1 (recognizing FG repeat nucleoporins [Snow et al, 1987]), mouse anti-nesprin-1 (8C3, gift from Dr. Glenn E. Morris, RJAH Orthopaedic Hospital, UK), mouse anti-nesprin-1 (7A12, #MABT843; Millipore), rabbit anti-nesprin-2 (#46-0705; Thermo Fisher Scientific, Invitrogen), rabbit anti-nesprin-3 (United States Biological Corporation), rabbit anti-myosin heavy chain (#124205; Abcam). DAPI (Sigma-Aldrich) were used to stain DNA.

## Western blotting

For Western blotting, cells were resuspended in 2X Laemmli buffer (4% SDS, 10% 2-mercaptoethanol, 20% glycerol, 0.004% bromophenol blue, and 0.125 M Tris–HCl pH 6.8) and boiled for 5 min.

Samples were run on a Novex Tris-Glycine gel (Life Technologies) using FASTRun Buffer (Thermo Fisher Scientific). Samples were then transferred to a nitrocellulose membrane (Life Technologies). Membranes were rinsed twice with TBS with 0.1% Tween-20 (Tw) and then blocked with 5% BSA in TBS/Tw. The membranes were incubated with primary antibody diluted in 0.5% BSA in TBS/Tw overnight at 4°C. The membranes were then washed six times with TBS/Tw and incubated with HRP-conjugated secondary antibodies in TBS/Tw for 1 h at room temperature. Signals were then developed using an enhanced chemiluminescence kit (Thermo Fisher Scientific) for 5 min before exposure to film.

### Immunofluorescence staining

For immunofluorescence staining, cells were plated on sterile glass coverslips the day before analysis. 24 h after plating, cells were rinsed with PBS containing calcium and magnesium and fixed using 2% PFA (#15710; Electron Microscopy Sciences) in PBS for 20 min. Samples were rinsed three times with PBS and blocked for 15 min using PBS with 5% goat serum (Jackson ImmunoResearch Laboratories) and 0.5% Triton X-100 (Tx) (Thermo Fisher Scientific). Samples were then incubated with primary antibody diluted in PBS with 1% goat serum and 0.1% Tx overnight at 4°C. After washing with PBS/0.1% Tx four times, samples were incubated with Alexa Fluor-conjugated secondary antibody diluted in PBS/0.1% Tx at room temperature for 1 h. Samples were finally washed twice with PBS/0.1% Tx, incubated with DAPI at room temperature for 10 min, and then washed twice with PBS and mounted on glass slides using Aqua-Poly Mount (Polysciences).

The proximity ligation assay was carried out as described previously (Cheng et al, 2022), using C3H cells that had been stably transduced with a lentiviral vector expressing Myc-lamin B1.

### Light microscopy and quantification

Confocal images were acquired on a Zeiss 780 or a Zeiss 880 Airyscan laser-scanning confocal microscope with a 63X PlanApo 1.4 NA objective. Contrast adjustment of the representative images was performed with ZEN software (Zeiss). 10 or more images from each stably or transiently transduced cell population of the lowest expression levels were randomly chosen and the NE/ER ratio was quantified. Lamin A staining was used to outline the nucleus and the areas of NE and ER were defined by −0.5 to 0 $\mu$m (NE) and +0.5 to +1 $\mu$m (ER) relative to the lamin A-defined nuclear edge using the "*Enlarge*" function in ImageJ (NIH). Total fluorescent intensities of V5 staining in both areas were measured and normalized to the calnexin staining of the same area. The ratio of NE/ER was then calculated by dividing the normalized V5 signals in the NE to the normalized V5 in the ER.

The co-localization analysis was performed with the "*Coloc2*" function in ImageJ. Where necessary, raw images were processed using the rolling ball "*background subtraction*" function in ImageJ. Control and test images were processed with identical parameters. Representative images were prepared with automatic Airyscan processing in ZEN.

## Data Availability

The proteomics datasets have been deposited in the public proteomics repository MassIVE (Mass Spectrometry Interactive Virtual Environment), part of the ProteomeXchange consortium (Vizcaino et al, 2014), with the identifier MSV000091154.

## Supplementary Information

## Acknowledgements

The authors thank Dr. Scott Henderson for advice on confocal microscopy, Titus Jung for assistance with proteomics data management, and Gerace and Yates laboratory members for helpful discussions. The project was supported by NIH grants U01DA040709 to L Gerace and P41 GM103533 to JR Yates.

### Author Contributions

L-C Cheng: data curation, formal analysis, supervision, investigation, visualization, and methodology.
X Zhang: data curation and investigation.
S Baboo: data curation and investigation.
JA Nguyen: resources and investigation.
S Martinez-Bartolomé: formal analysis.
E Loose: resources.
J Diedrich: investigation.
JR Yates III: resources, supervision, and funding acquisition.
L Gerace: conceptualization, supervision, funding acquisition, visualization, project administration, and writing—original draft, review, and editing.

### Conflict of Interest Statement

The authors declare that they have no conflict of interest.

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
