## [Reviewer comments · Life Science Alliance]

Life Science Alliance

Comparative membrane proteomics reveals diverse cell regulators concentrated at the nuclear envelope

Li-Chun Cheng, Xi Zhang, Sabyasachi Baboo, Julie Nguyen, Salvador Martinez-Bartolomé, Esther Loose, Jolene Diedrich, John R. Yates III, and Larry Gerace

DOI: <https://doi.org/10.26508/lsa.202301998>

Corresponding author(s): Larry Gerace, Scripps Research Institute and John R. Yates III, Department of Chemical Physiology California Campus

Review Timeline:

Submission Date:	2023-02-17
Editorial Decision:	2023-03-24
Revision Received:	2023-06-14
Editorial Decision:	2023-06-20
Revision Received:	2023-06-23
Accepted:	2023-06-26

Transaction Report:

March 24, 2023

Re: Life Science Alliance manuscript #LSA-2023-01998-T

Larry Gerace
Scripps Research Institute
Dept. of Cell Biology
The Scripps Research Institute
10666 N. Torrey Pines Road
La Jolla, USA-La Jolla, CA 92037 92037

Dear Dr. Gerace,

Thank you for submitting your manuscript entitled "Comparative membrane proteomics reveals diverse cell regulators concentrated at the nuclear envelope" to Life Science Alliance. The manuscript was assessed by expert reviewers, whose comments are appended to this letter. We invite you to submit a revised manuscript addressing the Reviewer comments.

Thank you for this interesting contribution to Life Science Alliance. We are looking forward to receiving your revised manuscript.

Sincerely,

B. MANUSCRIPT ORGANIZATION AND FORMATTING:

Reviewer #1 (Comments to the Authors (Required)):

It has become increasingly clear that nuclear envelope proteins play important roles in the cell and that defects in them are associated with a wide range of diseases. However, their identification can be challenging as many partition between the NE and the conjoined ER; thus it is necessary to determine the degree of enrichment of various proteins at the NE versus ER to discover proteins that play an important role at the former rather than being part of the background continuity of exchange of resident ER proteins or proteins being processed therein to be transported to other organelles, a process that is particularly challenging for less abundant proteins.

Here, a modified method is presented to identify these less abundant NE-enriched proteins by de Duvean co-fractionation analyses using quantitative comparisons between of membrane protein abundance by mass spectrometry for enriched nuclear membrane fractions and cytoplasmic membranes, candidates being evaluated by low-level expression to prevent oversaturation and spillover of otherwise bona fide NE proteins into the ER and other cytoplasmic membranes. Of many potentially interesting candidates that indeed enrich at the NE, one - Zdhhc6 - was functionally validated as a palmitoyltransferase for the INM-localized oxidoreductase Tmx4.

Overall this is an elegant study and an important contribution to our understanding of the organization and functions of the mammalian NE. The experiments are extremely carefully performed, controlled, and documented, and the results completely support the conclusions. I recommend publication.

- It's not absolutely clear how the MS data were normalized between samples and quantified. Presumably NSAF is Normalized Spectral Abundance Factor, but there are several ways of conducting such analyses and a better, more expanded description of how this was all performed would really help.
- The authors point out that "It is important to note that NE enrichment scores are meaningful only for proteins containing a TM domain(s)." What about those with membrane binding motifs (e.g. amphipathic helices) or similar? These are emerging as an important class of membrane-associate proteins. Presumably with the urea extraction, plus the TM domain prediction filter, these would be missed? It might be worth adding a cautionary note if so. Could one have used the solubilized chromatin fraction to exclude intranuclear proteins contaminating the NE (as they would be relatively depleted in the NE fraction), and thereby identified such specifically-associating NE candidates?
- There are missing y-axis titles in Figure S3C, S4B, S7B.

Reviewer #2 (Comments to the Authors (Required)):

The studies described in the manuscript by Cheng et al. used murine mesenchymal stem cells to look for low abundance NE proteins by biochemical enrichment of NE membranes and select pools of non-NE membranes for subtractive analysis. Comparative label-free MS proteomics was utilized to identify candidate NE enriched proteins, approximately a dozen of which were then validated by IF microscopy and found to be variably enriched in the NE (about 50% were obviously enriched versus simply present on the NE). Finally, one of the newly identified NE proteins, the palmitoyltransferase Zdhhc6, was functionally characterized by protein depletion coupled with MS isolation of palmitoylated proteins. The palmitoylation sites were identified on TMX4, one of the proteins likely palmitoylated directly by Zdhhc6, and shown to regulate TMX4 protein stability and NE enrichment. Overall, this is a well-prepared manuscript with a rigorous approach to further identify novel NE proteins, especially of low abundance. This would be of interest to the entire field of NE biologists as well as those that study any of the identified proteins.

Specific comments:

Fig 1B needs MW markers.

The anti-lamin A antibody description would suggest that it is instead an anti-lamin A/C antibody, given the amino acids it was raised against. If true, please rename, or if not please clarify what this antibody recognized and how it was made. I can't tell from the blot if there are two bands in the first lane or just one large one, which led to my looking at the antibody info.

Although the information is clear if the figure legends are carefully read, consider labeling figures to make them more readily interpretable. An example would be Fig S5 where RL1 could be labeled FG-repeat or NPC and V5 could be extended to V5-TMEM209, much like was done in Fig S4.

I am confused by something in Fig 4. In 4C I can evidence see that exogenous Lmna and TMX4 are palmitoylated. In 4D I can see that depletion of Zdhhc6 doesn't impact palmitoylation of Lmna but there is a decrease in palmitoylation of Tmx4 as seen with Strep-HRP. But the anti-V5 bands appear to be the same size, which doesn't match with the rest of the blots or, I suspect, reality. I assume that the anti-V5 is meant to show similar levels of V5 affinity capture between ciControl and siZdhhc6 samples for each protein. But I am left assuming that the image was inappropriately cropped and merged for the anti-V5 portion, something I can clearly see when I zoom in on the PDF. It would be appropriate to show the full portion of the blot, like was done in 4C, or if cropping must happen for some reason, it should be made clear that this is the case perhaps by drawing a box around each cropped section and including MW markers, although I would not recommend this approach.

Can you assess if lamin C is also palmitoylated? I would strongly suggest discussing the novelty and potential significance of lamin A palmitoylation, including any predicted site of modification in the protein.

Reviewer #3 (Comments to the Authors (Required)):

The manuscript entitled "Comparative membrane proteomics reveals diverse cell regulators concentrated at the nuclear envelope" by Cheng et al., has been submitted for publication in Life Science Alliance. This study employs an improved membrane proteomics approach to identify a group of previously unrecognized, low abundance transmembrane (TM) proteins that are enriched in the nuclear envelope (NE) but are not exclusively localized there. This is a carefully planned and executed study, the text is well written, and it makes a significant contribution to the field. I believe it should be published and I only have a few minor comments on the manuscript in its present form.

The main premise of the paper is the use of label-free proteomics comparing isolated NEs to an ensemble of all cytoplasmic organelle membranes. This contrasts with previous subtractive proteomics approaches that relied on ER-biased microsomal membranes and mostly led to the identification of abundant TM proteins that strongly accumulate at the NE under normal physiological conditions. The initial proteomics screen in the current study identified a large group of candidates assumed to be preferentially targeted to the NE relative to the peripheral ER. The next step included validation by low-level expression of the epitope-tagged candidates in C3H10T1/2 cells followed by quantification of the NE/ER localization ratio by immunofluorescence microscopy. Again, this represents improved methodology in respect to overexpressed constructs and the quantification is based on a comparison to endogenous calnexin, which is uniformly distributed in the NE/ER system. Ten out of sixteen proteins chosen for this validation step were found to preferentially associate with the NE and these include oxidoreductases, enzymes for lipid biosynthesis and regulators of cell growth and survival. This is a significant novel finding with interesting functional implications for future research on the NE. Finally, one of the validated NE-enriched TM proteins, Zdhhc6, was studied in more detail. Zdhhc6 is a palmitoyltransferase and the authors demonstrate that it modifies the previously identified, NE-localized, oxidoreductase Tmx4 (from a 2019 paper by the same authors). Interestingly, palmitoylation of Tmx4 by Zdhhc6 appears to reduce its NE localization. Although some details of the mechanism are still missing, this suggests that the interplay between these two proteins determines their dynamic localization and potential nuclear-specific functions.

Overall, this manuscript represents a significant step forward in proteomic studies of the NE and some interesting findings that will drive future functional studies. This is a long-standing effort led for many years by Larry Gerace and beginning with the seminal work of Schirmer et al., (Science, 2003). I doubt that anyone will attempt to reproduce the exact protocol used in this study, but nevertheless, the rationalization and detailed description are very important. Some of the findings (potential and validated candidates) are likely to serve as good starting points for new protein-focused studies and may interest scientists from additional fields.

Minor comments:

1. The authors should consider re-arranging the different parts of Figures 1 and 2 so that they fit the order of description in the text (pages 5-6), or vice versa. Although it's logical and relatively clear, the text jumps forward to Fig. 2 and the back to Fig. 1 and this is slightly confusing. I don't think any parts of the current figures should be moved to the supplemental information, but re-arrangement might help.
2. Ndc1 is considered to be a nuclear pore-membrane protein. The possible interpretation of its low NE-ES values (page 7) should be further discussed. Likewise, the potential localization of Tmem209 (pages 9-10) can be further elaborated upon in the Discussion.
3. An obvious point to the authors and most readers is that TM proteins are synthesized throughout the peripheral ER (e.g., top of page 12) and subsequently targeted to different cellular locations while undergoing PTMs. This complicates the interpretation of quantitative immunofluorescence. Although I don't doubt their conclusions, this should be spelled out more clearly in either of the relevant Results or Discussion sections.
4. Please fix typos:
 - Page 11, 2nd paragraph: "...which is best characterized target of"
 - Middle of page 13: "...inhibits apoptosis in multiple cells models"
 - Page 32, Figure 3 legend title: "...by ectopic targeting assay".

- awkward wording: top of page 13: "...at least many of these proteins are likely to act in"

Responses to Reviewers' Comments

Summary response:

We are genuinely grateful to the reviewers for examining our manuscript and for providing thoughtful comments that have enabled improvement. Additionally, we appreciate their positive perspectives on the work! We have responded to all comments with changes in the text and/or figures as indicated in detail below, and hope that these modifications adequately satisfy the recommendations.

Reviewer #1 (Comments to the Authors (Required)):

It has become increasingly clear that nuclear envelope proteins play important roles in the cell and that defects in them are associated with a wide range of diseases. However, their identification can be challenging as many partition between the NE and the conjoined ER; thus it is necessary to determine the degree of enrichment of various proteins at the NE versus ER to discover proteins that play an important role at the former rather than being part of the background continuity of exchange of resident ER proteins or proteins being processed therein to be transported to other organelles, a process that is particularly challenging for less abundant proteins.

Here, a modified method is presented to identify these less abundant NE-enriched proteins by de Duvean co-fractionation analyses using quantitative comparisons between of membrane protein abundance by mass spectrometry for enriched nuclear membrane fractions and cytoplasmic membranes, candidates being evaluated by low-level expression to prevent oversaturation and spillover of otherwise bona fide NE proteins into the ER and other cytoplasmic membranes. Of many potentially interesting candidates that indeed enrich at the NE, one - Zdhc6 - was functionally validated as a palmitoyltransferase for the INM-localized oxidoreductase Tmx4.

Overall this is an elegant study and an important contribution to our understanding of the organization and functions of the mammalian NE. The experiments are extremely carefully performed, controlled, and documented, and the results completely support the conclusions. I recommend publication.

- It's not absolutely clear how the MS data were normalized between samples and quantified. Presumably NSAF is Normalized Spectral Abundance Factor, but there are several ways of conducting such analyses and a better, more expanded description of how this was all performed would really help.

We now more explicitly describe the method used to determine Normalized Spectral Abundance Factor in the Material and Methods section (p 18):

"The Normalized Spectral Abundance Factor (NSAF) was used to estimate the relative abundance of individual proteins detected by MudPIT MS/MS (Zybailov et al, 2005). To determine NSAF values for each MudPIT run, the spectral count per unit (amino acid) length of all proteins detected was calculated, and this value for each protein was normalized to the summation of this value for all proteins identified in the same MudPIT run, thereby describing a normalized abundance of each protein. The NSAF values for each protein in the various membrane fractions analyzed were used to calculate NE enrichment scores."

- The authors point out that "It is important to note that NE enrichment scores are meaningful only for proteins containing a TM domain(s)." What about those with membrane binding motifs (e.g. amphipathic helices) or similar? These are emerging as an important class of membrane-associate proteins. Presumably with the urea extraction, plus the TM domain prediction filter, these would be missed? It might be worth adding a cautionary note if so.

This is an interesting point. We scanned our datasets for well-studied "non-TM" proteins with membrane binding domains or features (e.g. Hurley, BBA, 2006) that were detected in most experimental repeats. According to Uniprot, the most prominent non-TM proteins in our datasets with other membrane-association properties were Cav1 and Arf1, which contain both covalently bound lipid and amphipathic helices. Contradicting the Uniprot assignment, however, we found that Cav1 was confidently predicted to have a TM segment by the CCTOP server, which could explain its extraction resistance. The Arf1 enrichment in extracted membranes may be due to both its associated myristate and amphipathic helices, but also could reflect additional supramolecular interactions. Further scrutiny of our data revealed that covalently bound lipid and amphipathic helices alone were not sufficient to confer extraction resistance, as exemplified by lamin B1 and certain Nups (see below). Our overall assessment is that the presence of membrane association features in "non-TM" proteins, by themselves, doesn't afford predictive insight.

We have added the following sentences to the text to represent this issue (p 8) :

"In unusual cases, extraction-resistant behavior of membrane proteins might be dictated by protein features other than a TM domain, such as membrane binding motifs, covalently attached lipid and/or supramolecular associations (Hurley, 2006). However, these features by themselves are not predictive, since efficient membrane extraction was seen for lamin B1, which is stably modified by farnesylation (Burke & Stewart, 2013), and for the nucleoporins Nup160, Nup133, Nup155 and Nup35, which contain membrane-associating amphipathic helices (Hamed & Antonin, 2021) (Table S2)."

Could one have used the solubilized chromatin fraction to exclude intranuclear proteins contaminating the NE (as they would be relatively depleted in the NE fraction), and thereby identified such specifically-associating NE candidates?

Yes. We have highlighted this point by adding the following sentence to the text (p 8):

"Many proteins in this category were revealed by a previous analysis that included a "nuclear contents" fraction (Fig 1A) (Cheng et al, 2019), and usually are apparent from their annotations in Uniprot."

- There are missing y-axis titles in Figure S3C, S4B, S7B.

These Y-axis titles are now provided for the three figures noted.

Reviewer #2 (Comments to the Authors (Required)):

The studies described in the manuscript by Cheng et al. used murine mesenchymal stem cells to look for low abundance NE proteins by biochemical enrichment of NE membranes and select pools of non-NE membranes for subtractive analysis. Comparative label-free MS proteomics

was utilized to identify candidate NE enriched proteins, approximately a dozen of which were then validated by IF microscopy and found to be variably enriched in the NE (about 50% were obviously enriched versus simply present on the NE). Finally, one of the newly identified NE proteins, the palmitoyltransferase Zdhhc6, was functionally characterized by protein depletion coupled with MS isolation of palmitoylated proteins. The palmitoylation sites were identified on TMX4, one of the proteins likely palmitoylated directly by Zdhhc6, and shown to regulate TMX4 protein stability and NE enrichment. Overall, this is a well-prepared manuscript with a rigorous approach to further identify novel NE proteins, especially of low abundance. This would be of interest to the entire field of NE biologists as well as those that study any of the identified proteins.

Specific comments:

Fig 1B needs MW markers.

We have modified the former Fig. 1B (currently Fig. 2B; please see response to Reviewer #3, below) to include larger gel regions with 3-4 MW markers that are now labeled in the Figure. Since this modified Figure panel occupies considerably more space than the original one, we have retained only actin as a representative cytoskeletal marker, removing the two other cytoskeleton controls (vimentin and myosin heavy chain), whose fractionation can be more quantitatively followed in Table S2.

The anti-lamin A antibody description would suggest that it is instead an anti-lamin A/C antibody, given the amino acids it was raised against. If true, please rename, or if not please clarify what this antibody recognized and how it was made. I can't tell from the blot if there are two bands in the first lane or just one large one, which led to my looking at the antibody info.

The antibody was made to a synthetic peptide from a shared regions of lamins A/C. In the blot shown, lamins A/C are not resolved in the "prewash" NE sample, now noted in the Fig 2 legend. This is due to blot exposure needed to visualize lamin A in the carbonate and urea-washed samples. The provenance of the antibody is more clearly described in the Materials and Methods of the current manuscript (p 23):

"...rabbit anti-lamin A (affinity purified, made in-house to residues 396-429 of human lamin A and C) (Schirmer et al, 2001)."

Although the information is clear if the figure legends are carefully read, consider labeling figures to make them more readily interpretable. An example would be Fig S5 where RL1 could be labeled FG-repeat or NPC and V5 could be extended to V5-TMEM209, much like was done in Fig S4.

We now have done this in Figs. S3, S4, S5 and S6.

I am confused by something in Fig 4. In 4C I can evidence see that exogenous Lmna and TMX4 are palmitoylated. In 4D I can see that depletion of Zdhhc6 doesn't impact palmitoylation of Lmna but there is a decrease in palmitoylation of Tmx4 as seen with Strep-HRP. But the anti-V5 bands appear to be the same size, which doesn't match with the rest of the blots or, I suspect, reality. I assume that the anti-V5 is meant to show similar levels of V5 affinity capture between ciControl and siZdhhc6 samples for each protein. But I am left assuming that the image was inappropriately cropped and merged for the anti-V5 portion, something I can clearly see when I zoom in on the PDF. It would be appropriate to show the full portion of the blot, like was done in 4C, or if cropping must happen for some reason, it should be made clear that this is the case

perhaps by drawing a box around each cropped section and including MW markers, although I would not recommend this approach.

Good point. We have modified Fig 4C to show the entire blot, as previously shown in Fig 4B.

Can you assess if lamin C is also palmitoylated? I would strongly suggest discussing the novelty and potential significance of lamin A palmitoylation, including any predicted site of modification in the protein.

Since Lamin C is essentially a truncated form of lamin A, we were unable to detect it uniquely in the mass spectrometry palmitoylation analysis (Table S4). Also we didn't analyze potential palmitoylation of lamin C as a transiently transfected cDNA (Fig 4). However, the predicted sites of lamin A palmitoylation (below paragraph) are absent from lamin C.

Regarding the novelty and potential significance of lamin A palmitoylation, lamin A appeared in the palmitoylated protein database from several studies (summarized on the SwissPalm website) but its modification was not confirmed by the *de rigueur* method of direct labeling/SDS PAGE such as we have done in Fig 4C. The online PalmPred program predicts lamin A modification on Cys588 and/or Cys592, which are conserved among mammals, frogs, and zebrafish (but not in flies; worms have only lamin B). We have added the following sentences to the Results section (p 11) to reflect these findings:

“Previous screens have suggested that lamin A is palmitoylated (SwissPalm database), and the PalmPred program predicts palmitoylation of human lamin A on Cys588 and/or Cys592. Since these residues are conserved in mammals as well as in zebrafish, the functions of this modification merit further study.”

Reviewer #3 (Comments to the Authors (Required)):

The manuscript entitled "Comparative membrane proteomics reveals diverse cell regulators concentrated at the nuclear envelope" by Cheng et al., has been submitted for publication in Life Science Alliance. This study employs an improved membrane proteomics approach to identify a group of previously unrecognized, low abundance transmembrane (TM) proteins that are enriched in the nuclear envelope (NE) but are not exclusively localized there. This is a carefully planned and executed study, the text is well written, and it makes a significant contribution to the field. I believe it should be published and I only have a few minor comments on the manuscript in its present form.

The main premise of the paper is the use of label-free proteomics comparing isolated NEs to an ensemble of all cytoplasmic organelle membranes. This contrasts with previous subtractive proteomics approaches that relied on ER-biased microsomal membranes and mostly led to the identification of abundant TM proteins that strongly accumulate at the NE under normal physiological conditions. The initial proteomics screen in the current study identified a large group of candidates assumed to be preferentially targeted to the NE relative to the peripheral ER. The next step included validation by low-level expression of the epitope-tagged candidates in C3H10T1/2 cells followed by quantification of the NE/ER localization ratio by immunofluorescence microscopy. Again, this represents improved methodology in respect to overexpressed constructs and the quantification is based on a comparison to endogenous calnexin, which is uniformly distributed in the NE/ER system. Ten out of sixteen proteins chosen for this validation step were found to preferentially associate with the NE and these include oxidoreductases, enzymes for lipid biosynthesis and regulators of cell growth and survival. This is a significant novel finding with interesting functional implications for future research on the

NE. Finally, one of the validated NE-enriched TM proteins, Zdhhc6, was studied in more detail. Zdhhc6 is a palmitoyltransferase and the authors demonstrate that it modifies the previously identified, NE-localized, oxidoreductase Tmx4 (from a 2019 paper by the same authors). Interestingly, palmitoylation of Tmx4 by Zdhhc6 appears to reduce its NE localization. Although some details of the mechanism are still missing, this suggests that the interplay between these two proteins determines their dynamic localization and potential nuclear-specific functions. Overall, this manuscript represents a significant step forward in proteomic studies of the NE and some interesting findings that will drive future functional studies. This is a long-standing effort led for many years by Larry Gerace and beginning with the seminal work of Schirmer et al., (Science, 2003). I doubt that anyone will attempt to reproduce the exact protocol used in this study, but nevertheless, the rationalization and detailed description are very important. Some of the findings (potential and validated candidates) are likely to serve as good starting points for new protein-focused studies and may interest scientists from additional fields.

Minor comments:

1. The authors should consider re-arranging the different parts of Figures 1 and 2 so that they fit the order of description in the text (pages 5-6), or vice versa. Although it's logical and relatively clear, the text jumps forward to Fig. 2 and the back to Fig.1 and this is slightly confusing. I don't think any parts of the current figures should be moved to the supplemental information, but re-arrangement might help.

Thanks for raising this point. We were uncertain on how to present these data in the original manuscript. To expedite flow, we have switched the order of the previous Fig 1 and Fig 2, so that the scheme for subcellular fractionation (now Fig 1A) is presented at the very beginning to coincide with the text. We do believe that the panels in each of the individual figures are conceptually linked and are best retained in the current configuration. Although the data in the current Fig 1B is not discussed until after the data in Fig 2, we believe that this presentation is less distracting than previously.

2. Ndc1 is considered to be a nuclear pore-membrane protein. The possible interpretation of its low NE-ES values (page 7) should be further discussed. Likewise, the potential localization of Tmem209 (pages 9-10) can be further elaborated upon in the Discussion.

The basis for the low NE-ES values for Ndc1 remains a mystery, as high NE-ES values were obtained for the other two TM proteins associated with the NPC, i.e., Pom121 and Smpd4. This may be a feature of certain developmental lineages such as C3H and other mesenchymal cells, as we also found relatively low NE enrichment of Ndc1 in adipocytes differentiated from C3H cells (Cheng et al, 2019). In yeast, Ndc1 is known to function at the spindle pole body as well as at the NPC. We have re-written the paragraph containing the Ndc1 discussion (p 7) to incorporate these thoughts. Hopefully this will inspire a broader consideration of Ndc1 functions:

“For the NE markers, almost all proteins with the exception of Ndc1 had NE-ES1 values > 0.9. The NE-ES2 values for these proteins were similarly high, reflecting the minimal detection of these components in the HCM fractions. The relatively low NE-ES values for Ndc1 (0.75, NE-ES1; 0.53, NE-ES2) contrasted with the high values seen for two other NPC-associated TM proteins, Pom121 and Smpd4 (NE-ES1 and NE-ES2, 0.98-1; Table S3). These results, together with data from differentiated adipocytes (Cheng et al., 2019), suggest that there may be a substantial pool of Ndc1 in the peripheral ER in certain cell types. Hypothetically, this pool could be engaged in functions other than NPC organization. In yeast, for example, Ndc1 has functions at the spindle pole body as well as at the NPC (Knockenbauer & Schwartz, 2016; Lin & Hoelz, 2019). In a related situation, a peripheral ER pool has been reported for emerlin in

certain cell types and conditions, and functions for emerin outside the INM have been described (Berk et al, 2013; Le et al, 2016)."

We have expanded our discussion of Tmem209, and moved this to the discussion (p 12):

"Interestingly, we found that ectopically expressed Tmem209 yielded punctate immunofluorescence labeling at the NE that strongly co-localized with NPCs. It was the only protein examined in the ectopic targeting assay with this characteristic. Moreover, Tmem209 has a plant ortholog that is suggested to be NPC-associated (Goto et al, 2019; Tang et al, 2020). Since its abundance in C3H cells is similar to that of the TM nucleoporins Pom121 and Ndc1 (Table S1), Tmem209 may be a previously unrecognized architectural component of the NPC in animals that merits further analysis."

3. An obvious point to the authors and most readers is that TM proteins are synthesized throughout the peripheral ER (e.g., top of page 12) and subsequently targeted to different cellular locations while undergoing PTMs. This complicates the interpretation of quantitative immunofluorescence. Although I don't doubt their conclusions, this should be spelled out more clearly in either of the relevant Results or Discussion sections.

We have added the following sentence to the Results section to promote deeper thinking about the limitations of our assay, at the point where we describe the ectopic targeting assay (p):

"Whereas this ectopic targeting assay provides an accessible proxy for reporting the distribution of the endogenous proteins, the results reflect a composite of the rates of synthesis (in the peripheral ER) and NE targeting and turnover (in the two compartments), which might differ somewhat between endogenous proteins and ectopic counterparts."

4. Please fix typos:

- Page 11, 2nd paragraph: "...which is best characterized target of"
- Middle of page 13: "...inhibits apoptosis in multiple cells models"
- Page 32, Figure 3 legend title: "...by ectopic targeting assay".
- awkward wording: top of page 13: "...at least many of these proteins are likely to act in"

Thanks for the careful eye; we have fixed these.

June 20, 2023

RE: Life Science Alliance Manuscript #LSA-2023-01998-TR

Dr. Larry Gerace
Scripps Research Institute
Dept. of Molecular Medicine
The Scripps Research Institute
10550 N. Torrey Pines Road
La Jolla, USA-La Jolla, CA 92037 92037

Dear Dr. Gerace,

Thank you for submitting your revised manuscript entitled "Comparative membrane proteomics reveals diverse cell regulators concentrated at the nuclear envelope". We would be happy to publish your paper in Life Science Alliance pending final revisions necessary to meet our formatting guidelines.

- please be aware that LSA permits supplementary tables but not EV Tables. It is important only to upload tables that are intended for publication.
- please add ORCID ID for the secondary corresponding author--they should have received instructions on how to do so
- please be sure that all co-authors have been added to the author contribution section
- please add callouts for Figures S3A-C, S4A-B, S7A-B to your main manuscript text

A. FINAL FILES:

B. MANUSCRIPT ORGANIZATION AND FORMATTING:

**Submission of a paper that does not conform to Life Science Alliance guidelines will delay the acceptance of your

manuscript.**

The license to publish form must be signed before your manuscript can be sent to production. A link to the electronic license to publish form will be sent to the corresponding author only. Please take a moment to check your funder requirements.

Sincerely,

June 26, 2023

RE: Life Science Alliance Manuscript #LSA-2023-01998-TRR

Dr. Larry Gerace
Scripps Research Institute
Dept. of Molecular Medicine
The Scripps Research Institute
10550 N. Torrey Pines Road
La Jolla, USA-La Jolla, CA 92037 92037

Dear Dr. Gerace,

Thank you for submitting your Resource entitled "Comparative membrane proteomics reveals diverse cell regulators concentrated at the nuclear envelope". It is a pleasure to let you know that your manuscript is now accepted for publication in Life Science Alliance. Congratulations on this interesting work.

DISTRIBUTION OF MATERIALS:

Again, congratulations on a very nice paper. I hope you found the review process to be constructive and are pleased with how the manuscript was handled editorially. We look forward to future exciting submissions from your lab.

Sincerely,
